Article 

# A molecular mechanosensor for real-time visualization of appressorium membrane tension in *Magnaporthe oryzae*

Lauren S. Ryder [1], Sergio G. Lopez[2,4], Lucile Michels[3,4], Alice B. Eseola [1], Joris Sprakel [3], Weibin Ma [1] & Nicholas J. Talbot [1]✉

The rice blast fungus *Magnaporthe oryzae* uses a pressurized infection cell called an appressorium to drive a rigid penetration peg through the leaf cuticle. The vast internal pressure of an appressorium is very challenging to investigate, leaving our understanding of the cellular mechanics of plant infection incomplete. Here, using fluorescence lifetime imaging of a membrane-targeting molecular mechanoprobe, we quantify changes in membrane tension in *M. oryzae*. We show that extreme pressure in the appressorium leads to large-scale spatial heterogeneities in membrane mechanics, much greater than those observed in any cell type previously. By contrast, non-pathogenic melanin-deficient mutants, exhibit low spatially homogeneous membrane tension. The sensor kinase Δ*sln1* mutant displays significantly higher membrane tension during inflation of the appressorium, providing evidence that Sln1 controls turgor throughout plant infection. This non-invasive, live cell imaging technique therefore provides new insight into the enormous invasive forces deployed by pathogenic fungi to invade their hosts, offering the potential for new disease intervention strategies.

Many plant pathogens use specialized infection cells called appressoria to infect their hosts[1–3]. Appressoria facilitate pathogen entry into host tissue to cause disease, and the famous 'gold leaf' experiment demonstrated the capacity of some fungal appressoria to puncture the leaf surface using force generation rather than enzymatic activity[2,4]. Appressoria of the rice blast fungus *Magnaporthe oryzae* (synonym of *Pyricularia oryzae*)[5]—a major threat to global food security[6–8]—breach the tough surface of rice leaves and, remarkably, other hard synthetic surfaces by generating turgor of up to 8.0 MPa (~40 times the pressure of a car tyre[9]). This generates force at the base of the appressorium, measured in a related pathogen *Colletotrichum graminicola* using an optical waveguide, of 17 μN (ref. 10). By contrast, pressures measured in fungal or oomycete hyphae seldom exceed 0.8 MPa (ref. 11). *M. oryzae* appressoria have a melanin-lined cell wall impermeable to glycerol, but freely permeable to water, which rapidly enters the cell,

generating hydrostatic pressure. Mutation of the melanin biosynthetic enzyme-encoding genes *ALB1*, *RSY1* and *BUF1* causes loss of appressorium melanization, which enables movement of solutes and water through the appressorium cell wall, leading to loss of turgor generation and the ability to cause disease[12,13].

Directly measuring appressorium turgor has been challenging because of the enormous pressure generated, which preclude the use of pressure probes. Instead, researchers have used indirect methods such as the incipient cytorrhysis assay, which records the rate of cell collapse when appressoria are incubated in hyperosmotic solutions[9,14,15]. However, melanin-deficient mutants undergo plasmolysis rather than cell collapse when exposed to high concentrations of glycerol limiting the use of this assay[9,14]. The Flipper-TR probe, which contains a membrane-targeted twisted push–pull fluorophore is sensitive to mechanical forces acting on the plasma membrane. Previous reports

[1]The Sainsbury Laboratory, University of East Anglia, Norwich Research Park, Norwich, UK. [2]Cell and Developmental Biology, The John Innes Centre, Norwich Research Park, Norwich, UK. [3]Laboratory of Biochemistry, Wageningen University & Research, Wageningen, the Netherlands. [4]These authors contributed equally: Sergio G. Lopez, Lucile Michels. ✉e-mail: nick.talbot@tsl.ac.uk

have suggested the fluorescence lifetime of the probe changes linearly with plasma membrane tension in both yeast and mammalian cells[16]. In *M. oryzae*, the probe has been used to measure plasma membrane tension in hyphae of Guy11 and a *Δvast1* mutant, which affects TOR[17] signalling, and is implicated in the cAMP response, cell integrity and control of autophagy[18–22]. But, while the probe suggests that the *Δvast1* mutant has increased tension, experiments were performed only in hyphae[23].

Here we set out to explore whether we could visualize turgor generation in appressoria directly. Recently, a set of chemically modified molecular rotors were developed to yield complete microviscosity maps of cells and tissues in the cytosol, vacuole, plasma membrane and wall of plant cells[24]. These boron-dipyrromethene-based molecular rotors are rigidochromic, meaning their fluorescence lifetime depends on the mechanics of their surroundings, such as viscosity or membrane tension. The N⁺-BDP plasma membrane probe has, for instance, revealed differences in membrane mechanics between the plant root cap and the meristem. Fluorescence lifetime imaging microscopy (FLIM) revealed the plant meristem to undergo continuous growth and cell division, resulting in constant tension in the plasma membrane[24]. The tension increases spacing between lipids, leading to significant reduction in membrane rotor lifetime compared to the relaxed plasma membranes of root cap cells[25]. Furthermore, closer examination of the plasma membrane revealed distinct lipid microdomains within a single bilayer. Likewise, in root hairs the fluorescence lifetime was lower at the growing tip (3.6 ± 0.8 ns), compared with the non-growing epidermal cell plasma membrane (4.3 ± 0.6 ns). The change in lifetime corresponds to the increase in tension in the growing root hair tip, where membrane curvature is greatest. Plasmolysis assays in rotor-stained root hairs confirmed the probe's responsiveness to changes in tension within *Arabidopsis* root tissues, as the fluorescence lifetime within root hair tips significantly increased upon exposure to hyperosmotic stress and a drop in membrane tension.

In this Article, we report that the mechanosensor N⁺-BDP plasma membrane rotor probe can detect spatial variations in membrane tension in *M. oryzae* appressoria. We reveal how changes in microviscosity correlate with appressorium-mediated turgor-driven plant cell infection and show that, under the extreme pressures of an appressorium, the plasma membrane exhibits spatial heterogeneity in tension, a phenomenon not previously observed in living cells. We provide experimental validation of the requirement of melanin biosynthesis for appressorium turgor generation in *M. oryzae* and show that the Sln1 turgor sensor kinase[26,27] is necessary for controlling both the rate of turgor generation and its modulation before plant infection. The N⁺-BDP mechanosensor therefore provides a direct, quantitative measurement of the average tension at the appressorium membrane enabling new insight into the behaviour of eukaryotic cells under extreme pressure and the mechanobiology of plant infection by the blast fungus.

## Results

### N⁺-BDP reveals spatial variations in plasma membrane tension in *M. oryzae*

We first determined whether the mechanoprobe N⁺-BDP could reveal changes in appressorium-specific membrane tension during a time course of infection-related development of the wild-type *M. oryzae* strain Guy11. During initial stages of appressorium development, 4 h after conidia have germinated on hydrophobic glass coverslips, incipient appressoria are not melanized and have not yet generated turgor. By contrast, 24 h after inoculation, appressoria are mature and fully melanized, and generate high levels of turgor, deployed as mechanical force as appressoria are tightly bound to the hydrophobic surface, creating a tight seal necessary for appressorium function[8,28,29] (Fig. 1a). The chemical structure of the mechanoprobe N⁺-BDP is based on a modified phenyl-substituted boron-dipyrromethene molecular rotor, in which the phenyl ring carries an aliphatic tail with two permanent

cationic charges, creating a positive charge and thereby targeting the negatively charged phospholipid bilayer[24] (Fig. 1b). Upon staining, the probe is positioned between the tails of the bilayer, with its aliphatic tail facing towards phospholipid heads (Fig. 1c). Previous work using rotor-stained giant unilamellar vesicles (GUVs) composed of sphingomyelin, 1,2-dioleoyl-*sn*-glycero-3-phosphocholine and cholesterol (0.56:0.24:0.20) has enabled study of lipid phase transition. The lipid phase separation in GUVs creates an inhomogeneous biological membrane composed of different lipid microdomains, similar to formation of lipid microdomains in biological membranes by immiscibility of different lipids[24,30–32]. Upon staining different GUVs, the N⁺-BDP mechanoprobe demonstrated a stronger mechanical restriction for rotations imposed by the tightly packed and solid-like sphingomyelin-rich gel-like ordered phase, generating longer average fluorescence lifetimes compared with the less tightly packed liquid-like phase enriched in 1,2-dioleoyl-*sn*-glycero-3-phosphocholine. When considering an appressorium, we hypothesized that early-stage (4 h) appressoria would display a more compact membrane as a result of being under little or no tension, thereby causing mechanical restriction of the rotor probe upon photoexcitation and longer average fluorescence lifetimes (Fig. 1c). In a 24 h appressorium meanwhile exhibiting high appressorium turgor and high membrane tension, the membrane would become stretched and disordered, allowing free rotation of the probe and consequently shorter average fluorescence lifetimes (Fig. 1d). To test this hypothesis, we used the N⁺-BDP mechanoprobe to stain 4 h and 24 h appressoria of *M. oryzae* Guy11 to observe the spatial variations in membrane tension (Fig. 1e). Strikingly, when we selected the appressorium membrane for analysis (Extended Data Fig. 1) we observed that 4 h appressoria displayed a homogeneous high average rotor lifetime of 3.98 ± 0.084 ns (Fig. 1f and Supplementary Video 1), in contrast to mature appressoria (7.5–24 h) which displayed spatial heterogeneity and a significantly lower average rotor lifetime of 2.79 ± 0.026 ns (Fig. 1f and Supplementary Video 2). Intriguingly, in contrast to the low fluorescence lifetimes we consistently observed in appressoria, we observed consistent and uniform high fluorescence lifetimes in *M. oryzae* germ tubes (Extended Data Fig. 2). Considering the primary function of the germ tube is to deliver the contents of the conidium to the developing appressorium for maturation, these cells have no requirement for extreme turgor generation, which is corroborated by the rotor probe.

We next tested whether artificially lowering the turgor of appressoria by incubation in hyperosmotic concentrations of glycerol, would independently corroborate probe responsiveness to changes in membrane tension within an appressorium. Under hyperosmotic conditions, the fluorescence lifetime in a 24 h appressorium significantly increased from 2.79 ± 0.041 ns to 3.10 ± 0.067 ns upon addition of exogenous 1 M glycerol (Extended Data Fig. 3). This change is consistent with a drop in tension as water exits the appressorium by osmosis[13,26]. Considering melanin biosynthesis and deposition occur between 4 h and 8 h post-inoculation (hpi) on a glass coverslip[8], we reasoned that this would provide a suitable time to capture changes in local membrane tension and turgor. A real-time movie of a Guy11 appressorium stained with the N⁺-BDP rotor probe was captured during a 3 h period, in which spatial variations in membrane tension were apparent and the overall fluorescence lifetime decreased (Fig. 1g, Supplementary Video 3 and Extended Data Fig. 4).

Our initial analyses of turgor generation were carried out on unyielding glass coverslips, so we were interested in observing appressorium turgor on surfaces that can be penetrated. As the N⁺-BDP mechanoprobe stains all membranes it encounters, imaging appressoria on a leaf surface is technically challenging. We therefore inoculated sterile onion epidermis, pre-treated with chloroform before washing in sterile water (Extended Data Fig. 5a), which can be readily penetrated by the fungus. We found that appressorial membranes displayed consistent spatial heterogeneity (Extended Data Fig. 5b) and a very similar

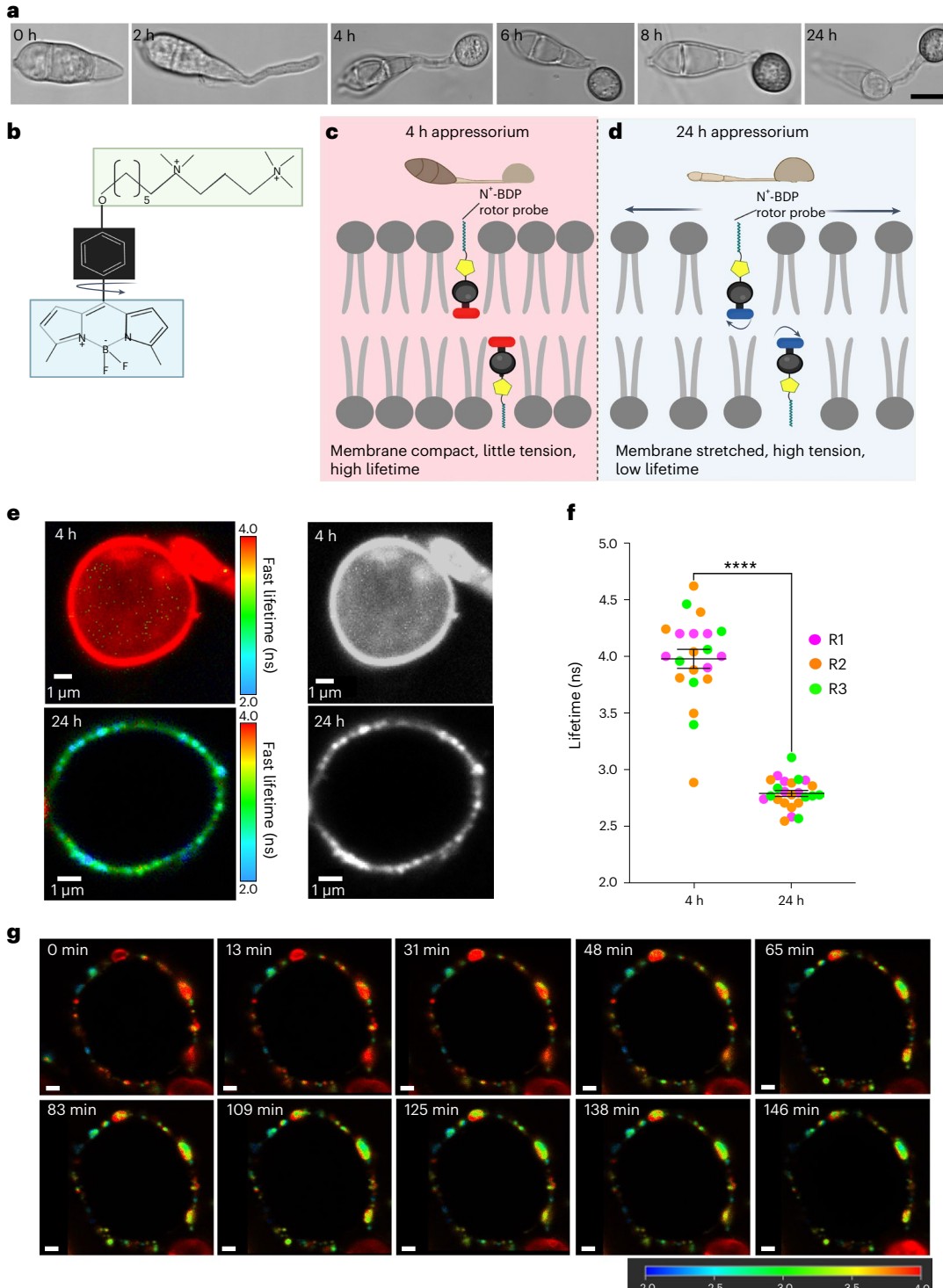

**Fig. 1 | Mechanosensor N+-BDP reveals spatial variations in membrane tension of appressoria of *M. oryzae*. a**, Time course of infection-related development of *M. oryzae*. Images show developing appressoria of the wild-type strain Guy11 germinated on glass coverslips between 0 h and 24 h. Images are representative of *n* = 3 independent repeats of the experiment. Scale bar, 10 μm. **b**, Chemical structure of the N+-BDP rotor. **c,d**, Schematic illustration showing the molecular mechanism by which N+-BDP reports changes in membrane tension in 4 h and 24 h Guy11 wild-type appressoria. **e**, Representative FLIM images of 4 h and 24 h N+-BDP rotor-stained appressoria. The colour corresponds to the fluorescence lifetime values expressed in nanoseconds, as shown in the key 2–4 ns. The corresponding black and white images highlight spatial

heterogeneity in membrane tension during turgor generation at 24 h. **f**, Dot plots showing the average fluorescence lifetime for 4 h and 24 h appressoria. Each dot corresponds to the average fluorescence lifetime obtained for an ROI drawn around the membrane of an individual appressorium in a 2D FLIM image. Total observations *n* = 45 appressoria examined in three biological replicates, typical range 21–24 appressoria; each biological replicate is colour coded (R1, replicate 1; R2, replicate 2; R3, replicate 3); data are presented as mean ± s.e.m. ****$P < 0.0001$, two-tailed unpaired Student's *t*-test with Welch correction. **g**, Time-lapse FLIM images of appressorium development in Guy11 4.5–7 hpi (0–145 min, respectively). Images are representative of *n* = 3 independent repeats of the experiment. Scale bars, 1 μm. Image created with BioRender.com.

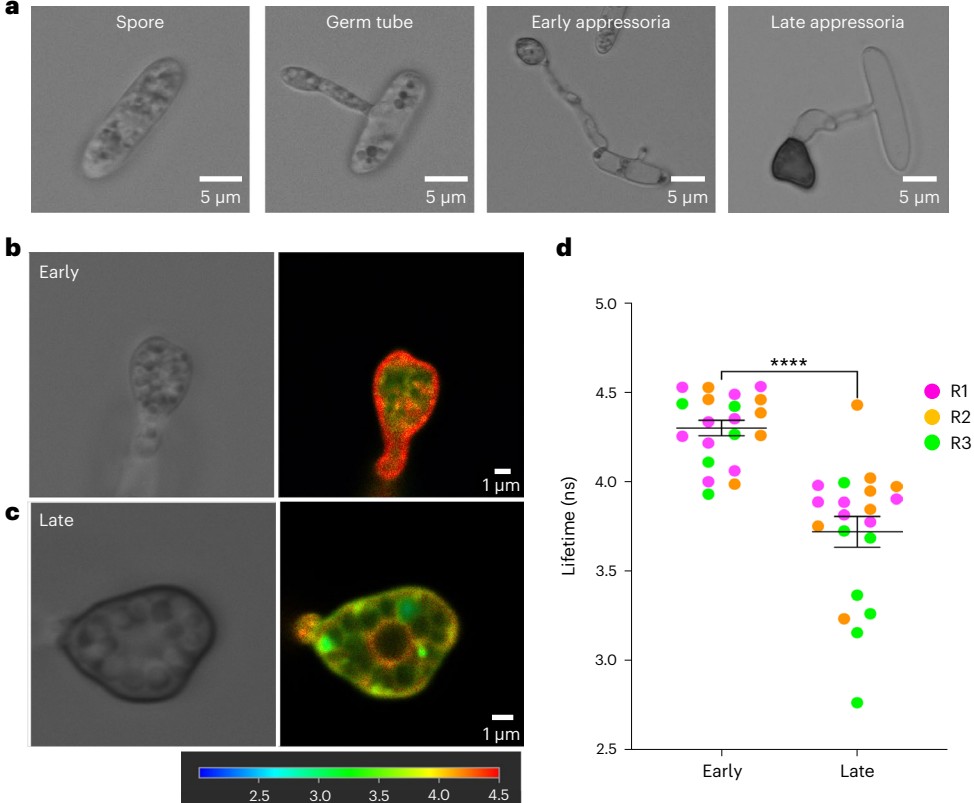

**Fig. 2 | Mechanosensor N⁺-BDP reveals variations in membrane tension of appressoria of the anthracnose pathogen *C. higginsianum*. a**, Time course of infection-related development of *C. higginsianum* development and maturation. Images show developing appressoria germinated on glass coverslips between 0 h and 24 h. Images are representative of *n = 3* independent repeats of the experiment. Scale bars, 5 μm. **b**, Representative FLIM images of early and late N⁺-BDP rotor-stained appressoria. The colour corresponds to the fluorescence lifetime values expressed in nanoseconds, as shown in the key 2.5–4.5 ns. **c**, Dot plots showing the average fluorescence lifetime for early and late appressoria.

Each dot corresponds to the average fluorescence lifetime obtained for an ROI drawn around the membrane of an individual appressorium in a 2D FLIM image. Total observations *n = 40* appressoria examined in three biological replicates; typical range 20 appressoria; each biological replicate is colour coded (R1, replicate 1; R2, replicate 2; R3, replicate 3); data are presented as mean ± s.e.m. ****$P < 0.0001$, **$P < 0.005$ two-tailed unpaired Student's *t*-test with Welch correction. Images are representative of *n = 3* independent repeats of the experiment. Scale bars, 1 μm.

---

lifetime compared with those on an unyielding coverslip surface at 2.69 ± 0.083 ns (Extended Data Fig. 5c). We conclude that the dynamics of appressorium turgor generation are similar on both yielding and unyielding surfaces.

To determine whether the N⁺-BDP mechanoprobe can be applied to other appressorium-forming fungal pathogens, we tested appressoria from the hemibiotrophic ascomycete fungus *Colletotrichum higginsianum*, which causes anthracnose disease on *Arabidopsis*[33]. During the initial stages of appressorium development, once conidia have germinated on hydrophobic glass coverslips, incipient appressoria of *C. higginsianum* are not melanized and have not yet generated turgor. By contrast, 24 h after inoculation, appressoria are mature and fully melanized (Fig. 2a). We used the N⁺-BDP mechanoprobe to stain early-stage (non-melanized) (Fig. 2b) and late-stage (melanized) (Fig. 2c) appressoria of *C. higginsianum*, to observe spatial variations in membrane tension. Early-stage appressoria displayed homogeneous high average rotor lifetime of 4.301 ± 0.043 ns (Fig. 2d), in contrast to late-stage appressoria which displayed a significantly lower average rotor lifetime of 3.72 ± 0.085 ns, consistent with our findings for *M. oryzae*. We conclude that the N⁺-BDP mechanoprobe can be widely deployed for generating membrane tension maps in appressorium-forming pathogens.

To confirm that the surprising large inhomogeneties observed in *M. oryzae* 24 h appressoria are due to changes in membrane tension and not the result of formation of compositional membrane microdomains

or dye interaction with the hydrophobic melanin layer, we performed control experiments using a plasma membrane sensor NR12S, a solvatochromic Nile red-based probe[34]. In the same way as the N⁺-BDP probe, the Nile red chromophore is functionalized with a lipid-like alkyl tail for membrane targeting (Extended Data Fig. 6a). This probe, however, is not sensitive to membrane tension, but rather exhibits a shift in emission wavelength in response to changes in local chemical polarity, such as lipid and sterol composition of the membrane. Ratiometric imaging, in which the total emission of the dye is split into two channels, provides a non-lifetime-based readout for the probe. Changes in membrane chemical composition and lipid phase affect the chemical polarity of the probe microenvironment, initiating a change in the intensity ratio between the blue and red channels[35] (Extended Data Fig. 6b,c). Previously, this probe was used for mapping spatial variation in plasma membrane chemical polarity of *Phytophthora infestans* germlings[35]. We observed that chemical polarity of appressoria becomes globally lower during appressorium maturation. In early 4 h appressoria, ratiometric imaging of the appressoria plasma membrane appears blue, indicating the membrane has a high chemical polarity that is is less ordered. (Extended Data Fig. 6d–j). However, in 24 h appressoria, the plasma membrane appears yellow and red, indicating the membrane has a low polarity indicative of a higher lipid packing order. (Extended Data Fig. 6k–p). Importantly, the magnified sections of the plasma membrane for both 4 h appressoria (Extended Data Fig. 6e,f,h,i) and 24 h appressoria (Extended Data Fig. 6l,m,o,p) shows the probe

NR12S displaying either homogeneous polarity, or polarity variations whose pattern is not consistent with the larger changes in tension we observe with the 4 h and 24 h appressoria stained with the N⁺-BDP rotor (for comparison, see Extended Data Fig. 6). We conclude that N⁺-BDP reveals spatial changes in appressorium-specific tension during infection-related development of *M. oryzae* and that the observed patterns are not caused by concurrent inhomogeneities in membrane composition.

### Melanin is critical for *M. oryzae* appressorium turgor

The synthesis of dihydroxy-naphthalene (DHN) melanin in *M. oryzae* has long been reported to be essential for appressorium-specific turgor-driven plant infection[7,13]. A layer of melanin is located between the appressorium membrane and cell wall where it acts as a structural barrier to efflux of solutes from the appressorium, essential for turgor generation and pathogenicity[7,13]. Mutation of genes encoding enzymes required for DHN-melanin biosynthesis, *ALB1*, *RSY1* and *BUF1*, causes impairment in appressorium and hyphal melanization[12]. Consequently, melanin-deficient mutants fail to infect intact host plants[12]. We determined whether N⁺-BDP could detect a reduction in membrane tension in *alb1⁻* and *buf1⁻* mutants compared with the isogenic wild-type Guy11. Both *alb1⁻* (Supplementary Video 4) and *buf1⁻* mutants displayed homogeneous high fluorescence lifetimes of $3.23 \pm 0.063$ ns and $3.20 \pm 0.056$ ns, respectively, similar to values for non-melanized 4 h appressoria in Guy11 (Fig. 3a,b,e). Furthermore, when Guy11 was exposed to the melanin biosynthesis inhibitor tricyclazole at 3 h compared to an untreated control, we observed a high fluorescence lifetime of $3.18 \pm 0.031$ ns, consistent with lifetimes of the melanin-deficient mutants and low appressorium tension (Fig. 3c–e). The mechanoprobe N⁺-BDP therefore demonstrates that *alb1⁻* and *buf1⁻* mutants do not generate appressorium turgor, and furthermore, that tension in the appressorial membrane of melanin mutants and tricyclazole-treated Guy11 is universally low. We also tested whether N⁺-BDP could detect an increase in membrane tension in non-melanized germ tube tips (Fig. 4a) during polarized growth compared with the wild type (Fig. 4b). Interestingly, the tips and points of curvature of germ tubes exhibited a lower fluorescence lifetime of $3.84 \pm 0.078$ ns compared with the subapical walls of the germ tube that displayed an average higher lifetime of $4.20 \pm 0.0773$ ns (Fig. 4b). Considering that melanin deposition has not begun at 3 hpi, with enzymes involved in melanin biosynthesis peaking in expression between 6 hpi and 8 hpi (*ALB1* MGG_07219, *RSY1* MGG_05059 and *BUF1* MGG_02252) (ref. 36) (Fig. 4c), this experiment effectively decouples melanization from membrane tension, showing that turgor generation in actively growing non-melanized hyphal tips can be revealed by the mechanosensor.

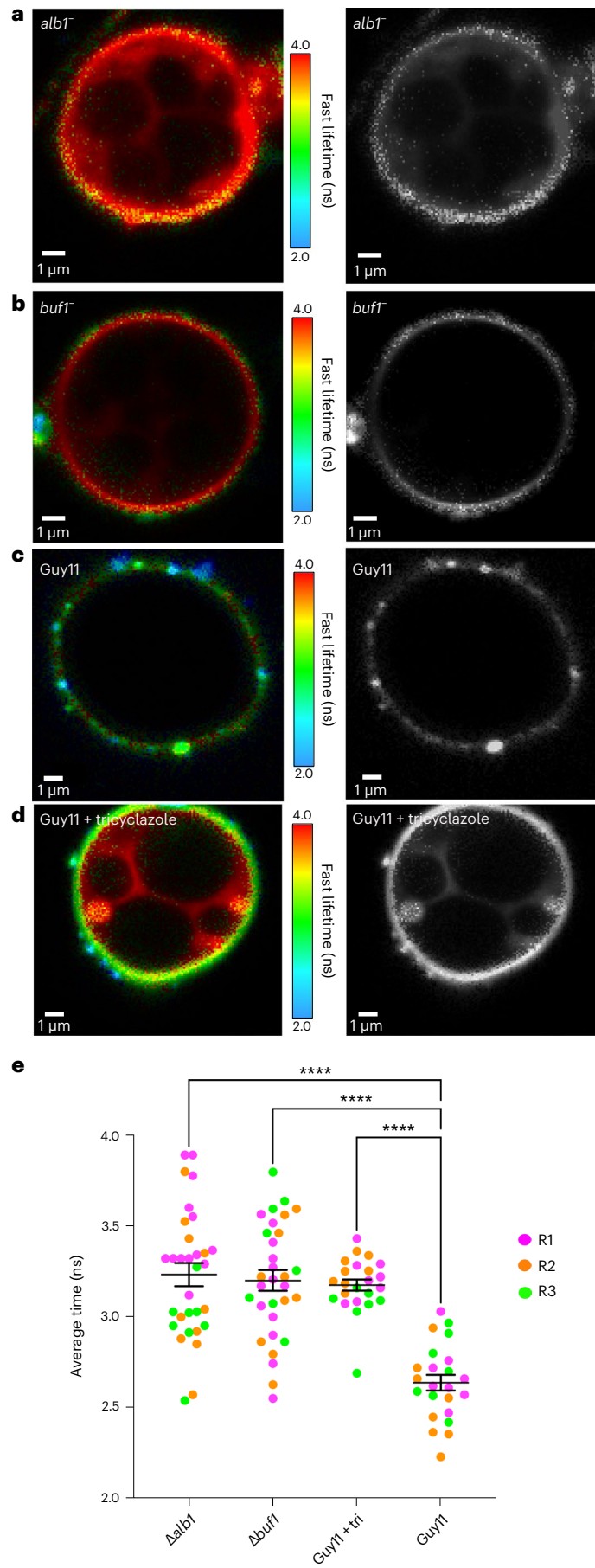

**Fig. 3 | The N⁺-BDP mechanosensor provides direct evidence for melanin-dependent appressorium turgor generation in *M. oryzae*. a**, FLIM image of an *alb1⁻* melanin-deficient mutant at 24 h germinated on glass coverslips and stained with the mechanosensory rotor probe N⁺-BDP. **b**, FLIM image of a *buf1⁻* melanin-deficient mutant at 24 h germinated on glass coverslips and stained with N⁺-BDP. **c**, FLIM image of Guy11 appressoria at 24 h germinated on glass coverslips and stained with N⁺-BDP. **d**, FLIM image of tricyclazole-treated appressoria of Guy11 at 24 h germinated on glass coverslips and stained with N⁺-BDP. The corresponding black and white images highlight membrane spatial heterogeneity/lack of spatial heterogeneity at 24 h. **e**, Dot plot showing the average fluorescence lifetime for *alb1⁻*, *buf1⁻*, Guy11+tricyclazole and Guy11 control appressoria imaged at 24 h. Each dot corresponds to the average fluorescence lifetime obtained for an ROI drawn around the membrane of an individual appressorium in a 2D FLIM image. Total observations $n = 108$ appressoria examined in three biological replicates; typical range 23–31 appressoria; each biological replicate is colour coded (R1, replicate 1; R2, replicate 2; R3, replicate 3); data are presented as mean ± s.e.m. ****$P < 0.0001$, as determined by one-way analysis (ANOVA) with Dunnett's multiple comparisons test. Images are representative of $n = 3$ independent repeats of the experiment. Scale bars, 1 µm.

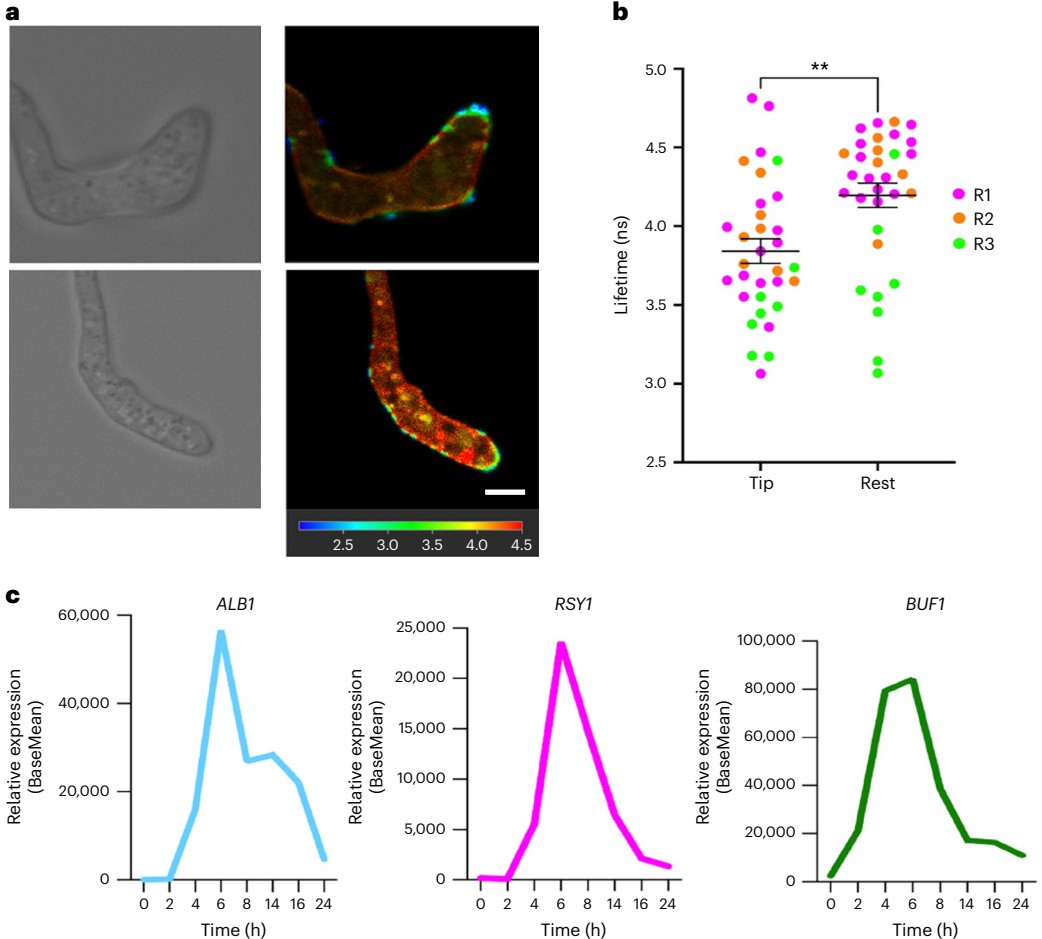

**Fig. 4 | The N⁺-BDP mechanosensor reveals spatial variations in membrane tension in developing germ tubes of *M. oryzae*. a**, FLIM micrographs of Guy11 at 3 hpi germinated on glass coverslips and stained with the rotor probe N⁺-BDP. Images are representative of *n* = 3 independent repeats of the experiment. **b**, Dot plots showing the average fluorescence lifetime for 3 h germ tubes. Each dot corresponds to the average fluorescence lifetime obtained for an ROI drawn around the membrane of an individual appressorium in a 2D FLIM image. Total observations *n* = 32 germ tubes in three biological replicates; each biological replicate is colour coded (R1, replicate 1; R2, replicate 2; R3, replicate 3); data are presented as mean ± s.e.m. **P < 0.002, two-tailed unpaired Student's *t*-test with Welch correction. **c**, Transcript abundance of genes that are involved in DHN-melanin biosynthesis during a time course of infection-related development (0–24 h) in Guy11. Gene expression is represented as base mean expression from *n* = 3 three RNA-seq experiments. Images are representative of *n* = 3 independent repeats of the experiment. Scale bars, 1 μm.

## Investigating the genetic control of appressorium turgor generation with N⁺-BDP

To test whether N⁺-BDP could provide new insight into turgor control in *M. oryzae*, we tested mutants impaired in appressorium function in which the effect on turgor generation is not known. Septins are required for pathogenicity of *M. oryzae*, regulating F-actin organization in the appressorium, and acting as lateral diffusion barriers for proteins involved in penetration peg emergence and elongation[37]. A total of six septins are present in *M. oryzae*, four of which share similarity to core septins identified in yeast, Sep3, Sep4, Sep5 and Sep6. Recently, very long chain fatty acid biosynthesis has been shown to regulate phosphatidylinositol-phosphate-mediated septin ring formation by recruiting septins to curved plasma membranes, initiating septin ring formation and subsequent penetration peg emergence[38]. Septin 5 is a core septin of *M. oryzae* that forms part of the heteropolymeric septin ring in the appressorium pore and is essential for plant infection[37]. We stained a *Δsep5* mutant with N⁺-BDP to see whether turgor generation is affected by absence of the septin. This revealed no significant change in membrane tension and appressorium turgor (2.90 ± 0.059 ns) compared with Guy11 (2.79 ± 0.046 ns) (Fig. 5a,b,d), providing evidence that absence of Sep5 has no effect on appressorium turgor generation,

but instead impairs re-polarization, separating the two processes and demonstrating more clearly the precise function of septins in *M. oryzae*[37]. We were also curious to test the *Δnox2* mutant, because previous work has shown that in the absence of *NOX2*, septins and F-actin do not assemble at the appressorium pore[39]. In addition to playing an important role in septin-mediated cytoskeletal re-organization, Nox enzymes are implicated in the chemiosmotic generation of turgor pressure, particularly in mammalian cells[40]. Staining the *Δnox2* mutant with the N⁺-BDP rotor probe revealed a significant reduction in membrane tension (3.12 ± 0.041 ns) compared with the wild-type Guy11 (2.79 ± 0.046 ns) (Fig. 5a,c,d), suggesting that absence of the Nox2 NADPH oxidase catalytic subunit does affect turgor generation in the appressorium. We conclude that the mechanoprobe N⁺-BDP is effective as a means of screening mutants impaired in appressorium function for a role in turgor pressure generation.

## The Sln1 kinase controls turgor throughout appressorium development

The Sln1 histidine-aspartate kinase in *M. oryzae* acts as a sensor to detect when a critical threshold of turgor has been reached in the appressorium to enable host penetration[26]. As a consequence *Δsln1*

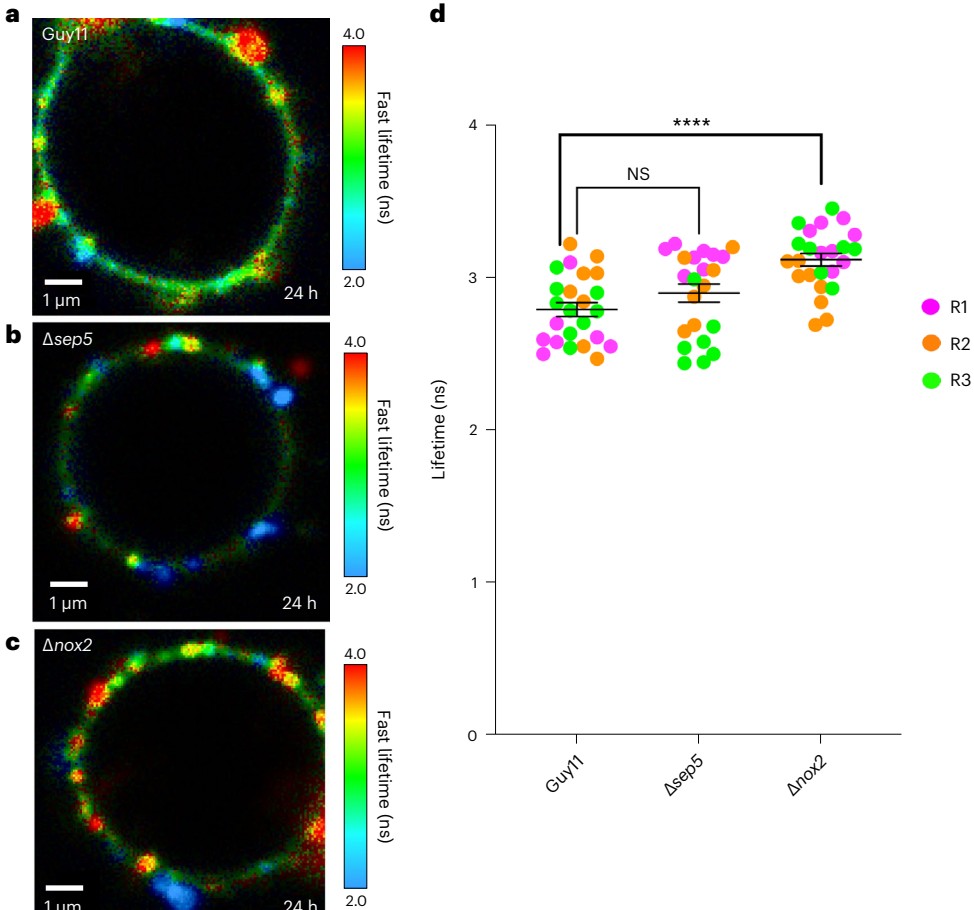

**Fig. 5 | N+-BDP identifies spatial variations in membrane tension in *M. oryzae* mutants impaired in appressorium function. a**, FLIM micrograph of Guy11 at 24 h germinated on glass coverslips and stained with the rotor probe N+-BDP. **b**, FLIM micrograph of an appressorium of the *Δsep5* mutant at 24 h germinated on glass coverslips and stained with N+-BDP. **c**, FLIM micrograph of an appressorium of a *Δnox2* mutant at 24 h germinated on glass coverslips and stained with N+-BDP. **d**, Dot plot showing the average fluorescence lifetime for Guy11 control, *Δsep5* and *Δnox2* appressoria imaged at 24 h. Each dot corresponds to the average fluorescence lifetime obtained for an ROI drawn around the membrane of an individual appressorium in a 2D FLIM image. Total observations *n* = 70 appressoria in three biological replicates; typical range 22–24 appressoria; each biological replicate is colour coded (R1, replicate 1; R2, replicate 2; R3, replicate 3); data are presented as mean ± s.e.m. Pairwise comparisons of fluorescence lifetime were made against Guy11 control ****P < 0.0001, two-tailed unpaired Student's *t*-test with Welch correction. Images are representative of *n* = 3 independent repeats of the experiment. Scale bar, 1 μm. NS, not significant.

mutants are unable to sense turgor and their appressoria are predicted to have excess turgor pressure and are hyper-melanized[26,27]. We therefore tested whether the N+-BDP mechanoprobe could detect aberrant turgor generation in a *Δsln1* mutant. First, we used septin–green fluorescent protein (GFP) localization to determine the time at which maximum turgor is achieved, when a septin ring is formed in the appressorium pore to facilitate re-polarization (Fig. 6a and Supplementary Video 5)[8,37,41]. F-actin and septin ring recruitment occurs in a pressure-dependent-manner[26,42] while in a melanin-deficient *buf1⁻* mutant[12], septin and F-actin localization are disordered[26,41,42]. The hyper-melanized *Δsln1* mutant also displays aberrant septin and actin localization patterns (Fig. 6c and Supplementary Video 6)[26]. To investigate whether the N+-BDP could detect the predicted abnormal turgor of *Δsln1* mutants we carried out staining of a time course of infection-related development and determined the average fluorescence lifetime at each developmental stage. In Guy11 appressoria at 4 hpi, an average lifetime of 3.95 ± 0.091 ns was observed, which significantly reduced to 3.11 ± 0.061 ns at 6 hpi, consistent with initiation of melanin synthesis and onset of turgor generation. By 8 hpi the average lifetime had significantly reduced again to 2.81 ± 0.079 ns, and by 24 hpi the average lifetime was 2.73 ± 0.042 ns. The average lifetime of N+-BDP fluorescence did not significantly

change between 8 hpi and 24 hpi, suggesting either that membrane tension remains constant after 8 hpi or that the rotor probe is saturated and unable to resolve higher tensions (Fig. 6b,e). The commitment point for septin ring organization is between 8 h and 10 h (refs. 8,37). Previously, septin ring formation was shown to be impaired after lowering appressorium turgor by application of exogenous glycerol, or treatment with tricyclazole when applied up to 16 hpi (ref. 26). This suggests that appressorium turgor reaches a critical threshold before septin ring assembly, and its modulation and maintenance through action of the Sln1-turgor-sensing-complex helps to stabilize conditions required for preserving septin ring organization. In the *Δsln1* mutant we observed significantly higher membrane tension particularly between 4 hpi and 6 hpi, averaging 3.70 ± 0.057 ns and 2.89 ± 0.110 ns, respectively. By 8 hpi the average lifetime significantly reduced to 2.55 ± 0.057 ns, and by 24 hpi the average lifetime observed was 2.51 ± 0.062 ns. Once again, the lifetime of the N+-BDP rotor probe remained constant between 8 hpi and 24 hpi, but the lifetime was significantly lower at 4 hpi, 8 hpi and 24 hpi compared with Guy11 (Fig. 6d,e and Supplementary Video 7). N+-BDP therefore reveals that Sln1 is necessary for controlling the temporal dynamics of turgor generation throughout appressorium development, because the *Δsln1* mutant exhibits excess turgor even at 4 hpi. In the current model for the action of Sln1, it is proposed to

act as a negative regulator of turgor once a threshold of turgor has been achieved. The model is therefore likely to be incorrect, or at least incomplete because it appears likely that the Sln1 kinase orchestrates turgor generation throughout appressorium development, controlling both the rate of turgor build-up and when sufficient pressure has been generated to facilitate the switch to polarized growth. Live cell imaging with the $N^+$-BDP rotor probe therefore reveals new insight into the regulation of appressorium turgor generation.

## Discussion

Many fungal pathogens develop infection cells to breach the tough external barrier of a plant or animal[1,2]. These cells include appressoria, hyphopodia and infection cushions[2,3,28,43–45]. Appressoria are, however, the most studied infection structure and essential for many of the most destructive plant diseases[46]. Economically important pathogens such as the powdery mildew pathogen *Blumeria graminis*, the corn smut fungus *Ustilago maydis* and soybean rust fungus *Phakopsora pachyrhizi*, for example, all elaborate appressoria. Oomycete pathogens, such as *Phytophthora* and *Pythium* species also develop functional appressoria[2], and recently it was shown that the late blight pathogen *Phytophthora infestans* enters its host at a diagonal angle, using a specific 'naifu' cutting action to break the host leaf surface[47].

The devastating rice blast fungus *M. oryzae* uses appressoria to breach the tough rice leaf cuticle, by generating enormous turgor of up to 8.0 MPa (ref. 13). In this study we applied a plasma membrane targeting rigidochromic molecular rotor to generate complete mechanical maps of wild-type and mutant appressoria of *M. oryzae*. These have revealed how the mechanics of the plasma membrane are adaptively modulated to accommodate appressorium growth and turgor generation. Creating detailed tension maps of appressoria during different stages of infection-related development has allowed us to observe real-time changes in turgor generation. Previous studies have suggested that changes in mechanical tension of a composite lipid membrane are facilitated through formation of bulges and protrusions of membrane domains[48]. In addition, other studies have suggested how mechanical stress on a membrane can increase the line tension between a microdomain and the rest of the lipid bilayer[49], which can in turn lead to microdomain growth[50]. How membranes deal with such extreme tensions, however, is unknown, because all membrane studies have been performed on cells with much lower internal pressures. By contrast, our study used fungal appressoria that generate pressures that are an order of magnitude higher than any previously analysed. This revealed that the appressorium membrane develops different microdomains when subject to extreme levels of mechanical stress. Control experiments with a probe sensitive to chemical polarity of membranes that can reveal compositional inhomogeneities, but that is tension insensitive, showed that the unexpected tension inhomogeneities do not correlate with formation of lipid microdomains or melanin binding of the probe. This is further corroborated by the fact that $N^+$-BDP can detect turgor changes in the tips of non-melanized hyphae.

Our study provides new information regarding how membrane tension is distributed under extreme turgor in eukaryotic cells, revealing much larger membrane spatial heterogeneities than previously reported. While previous studies in unwalled cells and free-standing in vitro membranes have linked tension to changes in membrane homogeneity[48,50], these are not pressurized to the same extent as membranes enclosed in appressoria. The findings presented here are therefore the first report of membrane heterogeneity under enormous pressures, and the results highlight how increases in tension can give rise to regimes in membrane physics, which are unknown and probably reflect the extreme mechanics of an appressorium. One potential explanation for the large tension inhomogeneities is that tension on the plasma membrane is very sensitive to the state of its anchoring to the cell wall in which it is enclosed, suggesting processes that couple wall and membrane mechanics. There are numerous proteins, including a variety of receptors, for example, that feature both transmembrane and extracellular domains and that mechanically link the membrane to the cell wall, potentially in a non-homogeneous manner that may be reflected in the tension patterns revealed.

To test the efficacy of the $N^+$-BDP mechanosensor, we first validated the well-known role of melanin in appressorium turgor generation. Here the rotor dye was able to reveal the severe impairment in appressorium turgor generation and the lack of membrane heterogeneity that accompanied reduced pressure of *M. oryzae* melanin-deficient mutants. We also showed that mutants in which turgor dynamics have not been investigated can be readily studied using $N^+$-BDP. While septin assembly is necessary for appressorium re-polarization and penetration peg emergence, our analysis revealed that a Δ*sep5* mutant did not show a significant reduction in turgor, based on FLIM analysis. This is consistent with previous reports that have shown that septin assembly is turgor dependent and occurs only once a threshold of pressure has been generated in the appressorium[26,41]. By contrast, the Δ*nox2* mutant showed a reduction in appressorium turgor. The Nox2 NAPDH oxidase catalytic subunit is necessary for appressorium function, including septin assembly and penetration peg formation[39]. Our analysis suggests that Nox2 may act upstream of septin assembly, serving a wider role in appressorium maturation than hitherto appreciated, including ensuring that sufficient turgor is generated. To investigate whether this is a direct result of inhibiting reactive oxygen species generation, chemical inhibition with ascorbic acid or the flavocytochrome inhibitor diphenylene iodonium could be carried out. It would also be valuable to investigate the function of the regulatory subunit NoxR in conditioning the ability of Nox2 to regulate appressorium turgor.

Finally, we tested whether $N^+$-BDP could reveal perturbations in appressorium turgor associated with the Δ*sln1* mutant[26]. Sln1 has been proposed to act as a turgor sensor kinase and is necessary to enable a mature appressorium to re-polarize and cause infection[26]. A mathematical model of appressorium-mediated plant infection predicted that a mutant lacking such a sensor would be unable to modulate appressorium turgor and therefore display excess pressure,

**Fig. 6 | $N^+$-BDP mechanosensor reveals that the Sln1 sensor kinase regulates turgor generation throughout appressorium development in *M. oryzae*.**
**a**, Time course of cortical septin ring formation during appressorium morphogenesis in *M. oryzae* wild-type strain Guy11. Conidial suspensions at $5 \times 10^4$ ml$^{-1}$ were inoculated onto glass coverslips and images captured at different time intervals during infection-related development (4–24 h). Scale bar, 5 μm. **b**, A FLIM time course of Guy11 appressorium development stained with the $N^+$-BDP probe (4–24 h). The colour translates the fluorescence lifetime values expressed in nanoseconds. Scale bar, 1 μm. **c**, Time course of cortical septin ring formation and mislocalization in a Δ*sln1* mutant during appressorium morphogenesis. Conidial suspensions at $5 \times 10^4$ ml$^{-1}$ were inoculated onto glass coverslips and images captured at different time intervals during infection-related development (4–24 h). Scale bar, 5 μm. **d**, A FLIM time course of Δ*sln1* appressorium development stained with $N^+$-BDP (4–24 h). **e**, Dot plots showing

the average fluorescence lifetimes for Guy11 and Δ*sln1* at 4 h, 6 h, 8 h and 24 h timepoints. Each dot corresponds to the average fluorescence lifetime obtained for an ROI drawn around the membrane of an individual appressorium in a 2D FLIM image. Pairwise comparisons were made between Guy11 timepoints, Δ*sln1* timepoints and like-for-like timepoints between the two strains. Total observations $n = 343$ appressoria in three biological replicates; typical range 19–27 appressoria; each biological replicate is colour coded (R1, replicate 1; R2, replicate 2; R3: replicate 3); data are presented as mean ± s.e.m. Guy11 4 h versus Guy11 6 h ****$P < 0.0001$, Guy11 6 h versus Guy11 8 h **$P < 0.005$, Δ*sln1* 4 h versus Δ*sln1* 6 h ****$P < 0.0001$, Δ*sln1* 6 h versus Δ*sln1* 8 h *$P < 0.01$, Guy11 4 h versus Δ*sln1* 4 h *$P < 0.02$, Guy11 8 h versus Δ*sln1* 8 h **$P < 0.008$, Guy11 24 h versus Δ*sln1* 24 h **$P < 0.005$ two-tailed unpaired Student's *t*-test with Welch correction. Images are representative of $n = 3$ independent repeats of the experiment. Scale bar, 1 μm. NS, not significant.

but would be unable to re-polarize[26]. The Δ*sln1* mutant displays these phenotypes, but until now its excess turgor was only predicted using the incipient cytorrhysis assay. Here we observed that Δ*sln1* mutants do show excess appressorium membrane tension revealed by the N⁺-BDP rotor. Even though we are clearly operating close to the limit of resolution of the rotor dye based on our calibration curve, because

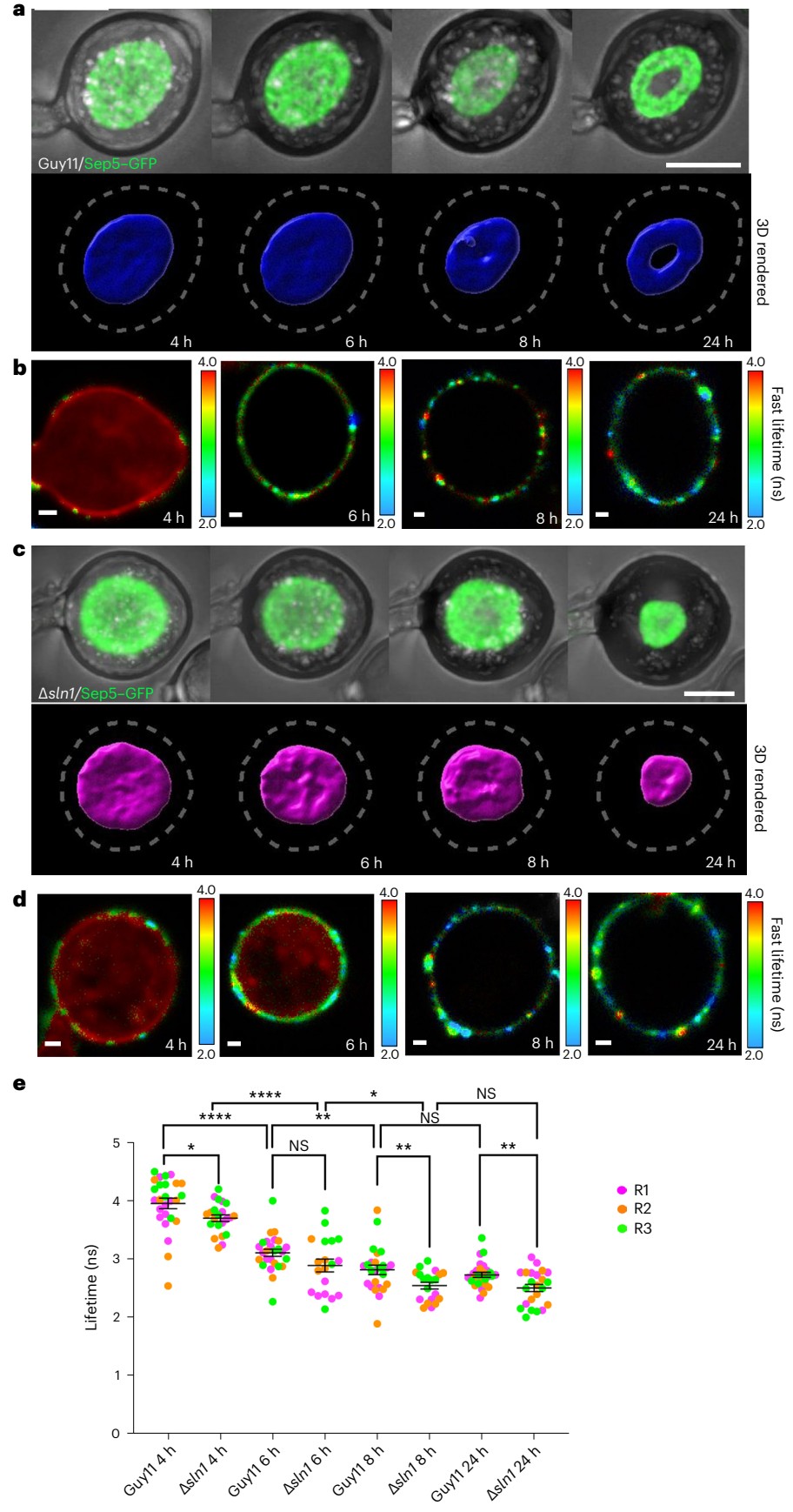

of the enormous pressures being measured, which are well beyond the scope of pressure probes for instance, a significant difference in turgor can be determined in mature appressoria of Δ*sln1* mutants compared with Guy11. This provides direct evidence that Sln1 does act as a sensor of appressorium turgor as predicted[26], but also shows that the kinase must act earlier than anticipated because it clearly modulates turgor during the initial stages of appressorium formation. Sln1 is therefore likely to control the rate of turgor generation, in addition to the switch to polarized growth and septin-dependent plant infection, playing a key role in the orchestration of plant infection. When considered together, these experiments highlight the utility of the N⁺-BDP mechanosensor as a direct means of analysing membrane tension in a living appressorium.

Fungal mechanobiology is a new and exciting approach for studying the mechanics of the plasma membrane, with scope to explore other compartments including the fungal cell wall, vacuoles and cytosol[24]. Future experiments employing the use of the molecular rotor may provide new quantifiable insights to spatial variations in microviscosity at the point of penetration, and at crossing points during cell-to-cell movement, where it is possible that transpressoria– which form specifically at cell junctions –generate turgor to successfully breach neighbouring cells[3,51]. Furthermore, a combination of surface-deformation imaging, rotor probe staining and mathematical modelling could help to establish the precise mode of entry and the forces translated at the host leaf surface which will prove invaluable in the search for effective blast disease control strategies[47].

## Methods

### Fungal strains and growth conditions

Growth and maintenance of *M. oryzae* was performed as described previously[52]. All strains used in the study are stored in the laboratory of N.J.T. and are freely available upon request.

### Two-dimensional FLIM imaging

Appressorium development was induced in vitro on borosilicate 18 × 22 mm glass coverslips (Thermo Fisher Scientific), as described previously[53]. For onion epidermis experiments, epidermal layers were incubated in chloroform for 30 min, washed five times in sterile dH₂O, before mounting onto a coverslip for inoculation. A 50 µl aliquot of conidial suspension ($5 \times 10^4$ ml⁻¹) was placed on a coverslip/onion epidermis and incubated at 24 °C. The aqueous phase of the droplet from Guy11 2–3 h germ tubes, 4 h, 24 h and *C. higginsianum* early and late appressoria was replaced with a 50 µl droplet of 10 µmol l⁻¹ N⁺-BDP probe dissolved in sterile distilled H₂O. Staining was performed at room temperature for 20 min for the hyper-melanized Δ*sln1* mutant and for 5 min for the wild-type Guy11 and all other mutants, after which unbound dye was removed by replacing 50 µl of the droplet five times with water. For calibration of the probe in appressoria, the aqueous phase of the droplet was removed and replaced with a 50 µl droplet of glycerol (0.2 M, 0.4 M, 0.75 M or 1 M). FLIM imaging was performed on a Leica TCS SP8X upright scanning confocal microscope coupled to a PicoHarp 300 TCSPC module (PicoQuant GmbH). Samples were excited with 488 nm output of a pulsed SuperK EXTREME supercontinuum white light laser (NKT Photonics) working at a repetition rate of 20 MHz. The full width at half maximum of the laser pulse was ~170 ps, as determined from instrument response functions recorded using Erythrosin B (Sigma-Aldrich, >95% purity) in KI-saturated (Sigma-Aldrich) water. Fluorescence emission was captured in the 510–530 nm range using a Leica HyD SMD detector. The objective lens was an HC Plan Apo 63×/ numerical aperture 1.20 water immersion objective (Leica Microsystems). SymPhoTime 64 (version 2.4, PicoQuant GmbH) was used to draw by hand in the 2D image a region of interest (ROI) that closely followed the membrane of the appressorium and did not include signal arising from any other part of the sample[54]. The same software was used to fit the overall fluorescence decay curve of the ROI with

a three-component exponential decay function. Fits were deemed acceptable only if residuals were evenly distributed around zero and the $\chi^2$ values were within 0.70–1.30 range. The average fluorescence lifetimes reported in this work are the intensity-weighted average lifetimes, which have been calculated as

$$< \tau >= \frac{\sum_i \alpha_i \tau_i^2}{\sum_i \alpha_i \tau_i}$$

where $a_i$ and $t_i$ are the amplitude and the lifetime of species *i*, respectively[55]. Images are reported in a false-colour scale that represents the weighted average of fluorescence decay for each pixel. For multi-exponential decays, the weighted average of fluorescence decay is equivalent to the intensity-weighted average fluorescence lifetime. Each datapoint in our analysis corresponds to the weighted average fluorescence lifetime obtained for the ROI of a single 2D image of an individual appressorium.

### FLIM time series experiments

Conidia were collected and inoculated onto glass coverslips. Early appressoria at 4 hpi were stained, washed and mounted onto glass slides, as previously described. FLIM imaging was performed on a Stellaris 8 Falcon upright scanning confocal microscope (Leica Microsystems). Samples were excited at 488 nm using a pulsed SuperK Fianium FIB-12 PP laser source (NK Photonics) working at a 20 MHz repetition rate. The full width at half maximum of the laser pulse was ~190 ps, as determined from instrument response functions obtained using Erythrosin B solution. Detection range and objective lens were the same as described above. A Leica HyD X detector was used and images acquired at 0 (4.5 h appressoria), 13, 31, 48, 65, 83, 109, 125, 138 and 146 min (7.5 h appressoria). The images were processed and analysed in LAS X (version 4.2, Leica Microsystems) and movies generated using Python distributed under the GNU public licence (Anaconda Ver. 3.8.10., https://anaconda.com). Python libraries used to generate the movie were NumPy[56], Scikit-image[57], pystackreg[58] and OpenCV[59]. The Python script can be found at https://github.com/SergioGabrielLopez/ movie_script.

The fluorescence lifetime for each frame was obtained by selecting an ROI and then fitting the overall fluorescence decay of the ROI with a four-exponential decay function. The fit was judged according to the previously mentioned criteria.

### Three-dimensional lifetime imaging

*M. oryzae* conidia were inoculated onto glass coverslips, stained with N⁺-BDP and imaged at desired times. Images were acquired on a Stellaris 8 FALCON upright scanning confocal microscope (Leica Microsystems). All imaging parameters were identical to those described for acquisition of the FLIM time series. The z-stacks had a length in the z-direction of ~12–15 mm, took 3–7 min to complete and were acquired in compliance with the Nyquist-Shannon sampling theorem. The 3D rendering of the z-stacks was carried out in LAS X (version 4.2, Leica Microsystems).

### 2D imaging of appressoria using NR12S chemical polarity probe

To image plasma membrane polarity in *M. oryzae* appressoria using the chemical polarity probe NR12S, a portion of the aqueous phase of the droplet (50 µl), was replaced with a solution of NR12S, dissolved at 10 µmol l⁻¹ in water. Staining was performed for 7 min, after which any unbound dye was removed by replacing 50 µl of the droplet five times with water. Two-dimensional ratiometric imaging with NR12S was performed on a Leica TCS SP8X upright scanning confocal microscope. Samples were excited with 514 nm output of a SuperK EXTREME supercontinuum white light laser (NKT Photonics) working at repetition rate of 80 MHz. Fluorescence was detected at 529–585 ('blue channel') and 610–700 nm ('red channel') using Leica HyD SMD detectors.

Ratiometric images obtained with NR12S staining were constructed from recorded intensity images using a custom MATLAB routine that divides photon count in each pixel of the blue channel image, by photon count in the corresponding pixel in the red channel image. Resulting images are reported in a false-colour scale that represents the intensity ratio for each pixel.

## Reporting summary

Further information on research design is available in the Nature Portfolio Reporting Summary linked to this article.

## Data availability

The *M. oryzae* genome database used in this study was http://fungi.ensembl.org/Magnaporthe_oryzae/Info/Index. All *M. oryzae* strains used in this study are freely available upon request from the corresponding authors. Detailed protocols for synthesis of both the N⁺-BDP rotor probe and the chemical polarity probe NR125 are available on request from J.S. Source data are provided with this paper.

## Code availability

The Python script used to produce the *M. oryzae* FLIM movie has been publicly deposited in GitHub at https://github.com/SergioGabrielLopez/movie_script.

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

## Acknowledgements

This project was funded by the BBSRC grant BB/V016342/1 and by the Gatsby Charitable Foundation. We thank C. Faulkner for providing us with *C. higginsianum* strain IMI 349061. We thank C. Wood for critical reading of the manuscript.

## Author contributions

L.S.R, N.J.T. and J.S. conceptualized the project. Experimental analyses were carried out by L.S.R, S.G.L, L.M, A.B.E and W.M. The paper was written by L.S.R, J.S and N.J.T.

## Competing interests

The authors declare no competing interests.

## Additional information

**Extended data** is available for this paper at https://doi.org/10.1038/s41564-023-01430-x.

**Correspondence and requests for materials** should be addressed to Nicholas J. Talbot.

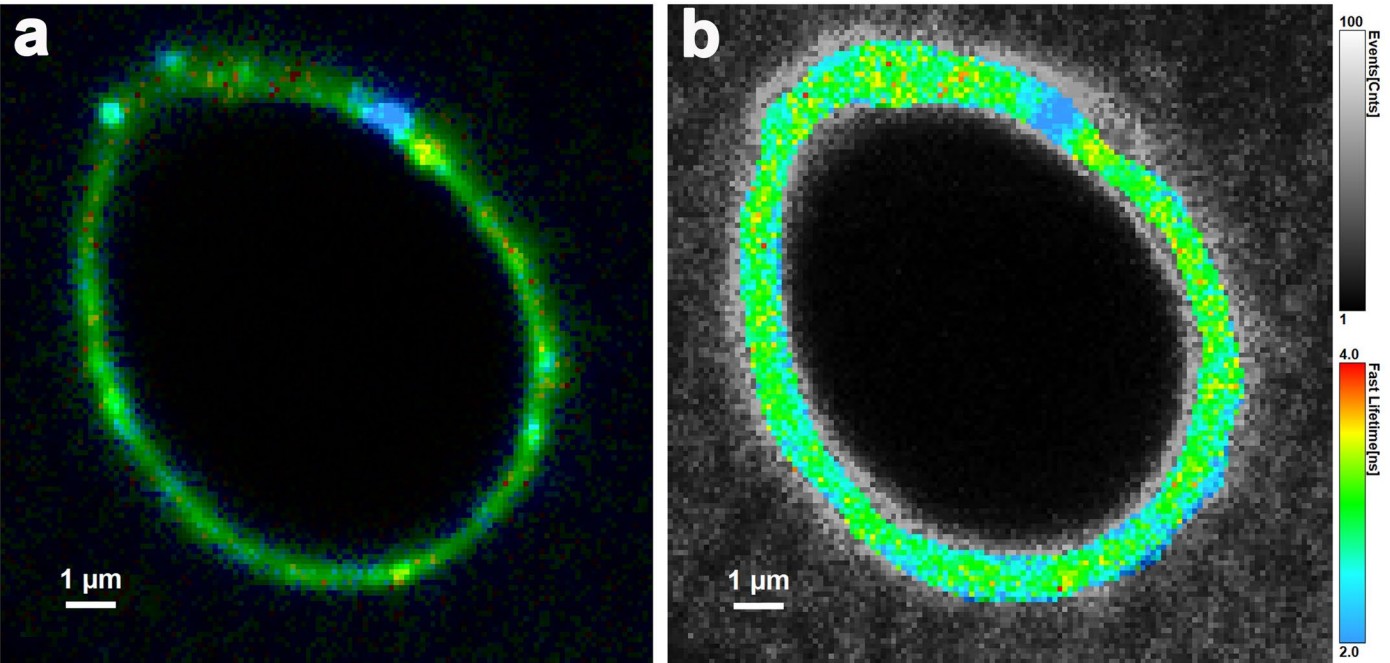

**Extended Data Fig. 1 | Region of interest selection in Two Dimensional FLIM images. a**, Representative FLIM image of a wild type Guy11 appressorium stained with the N⁺-BDP probe at 24 h. **b**, Image a showing the region of interest in colour with all other pixels in grayscale.

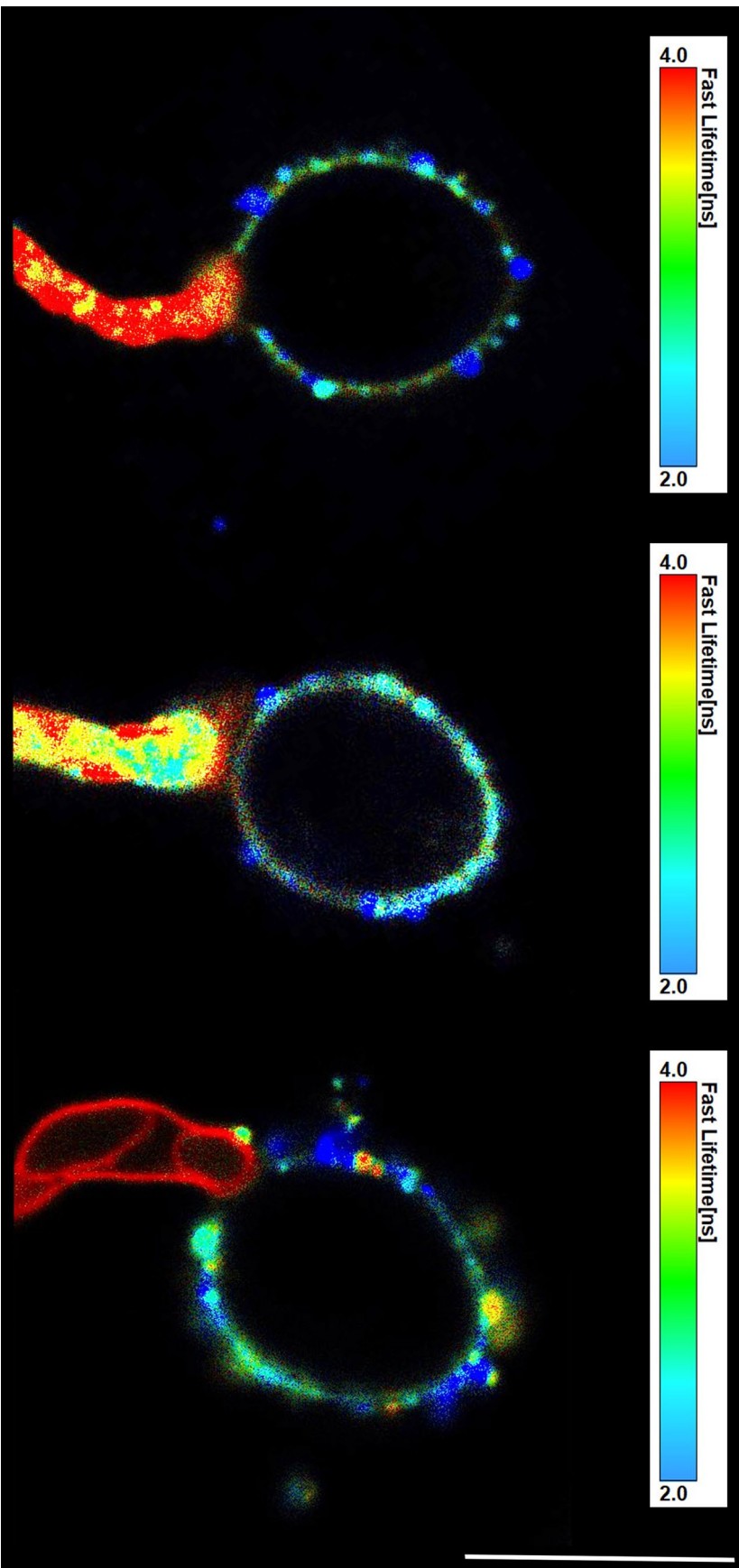

**Extended Data Fig. 2 | Spatial variations in membrane tension in germ tubes and appressoria of *M. oryzae*.** Representative FLIM images of wild type Guy11 24 h rotor stained appressoria and germ tubes. Images are representative of *n = 3* independent repeats of the experiment. Scale bar = 10 µm.

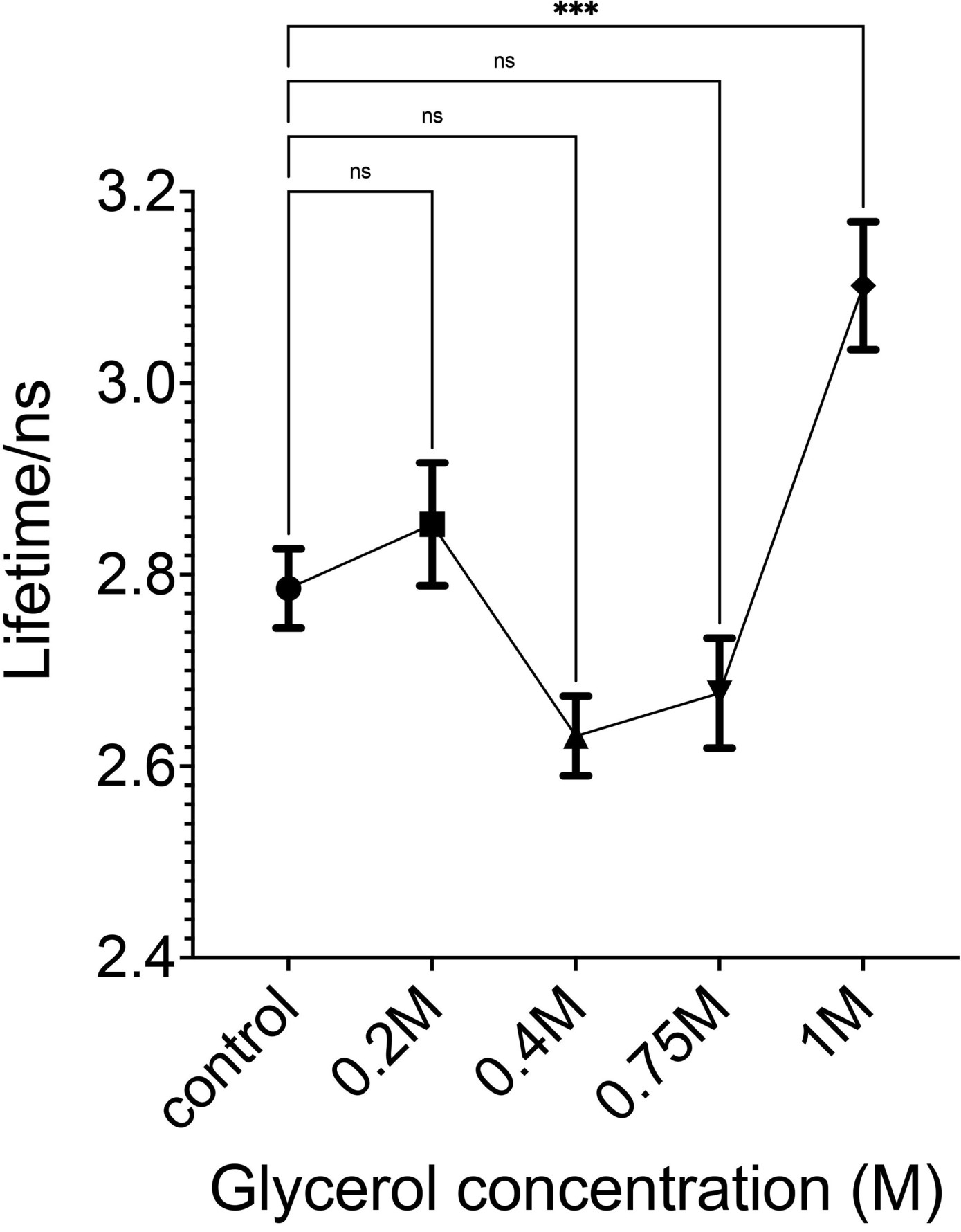

**Extended Data Fig. 3 | See next page for caption.**

**Extended Data Fig. 3 | Rotor dye N$^+$-BDP calibration in *M. oryzae* appressoria incubated in glycerol.** Line graph showing the average fluorescence lifetime of N$^+$-BDP stained appressoria of Guy11 incubated in different molar concentrations of glycerol. Values are means ± SEM for 3 biological replicates of the experiment, total observations $n = 129$ appressoria in 3 biological replicates; typical range 22–34 appressoria; data are presented as mean ± SEM. ***$P < 0.0002$ as determined by one-way analysis (ANOVA) with Dunnett's multiple comparisons test.

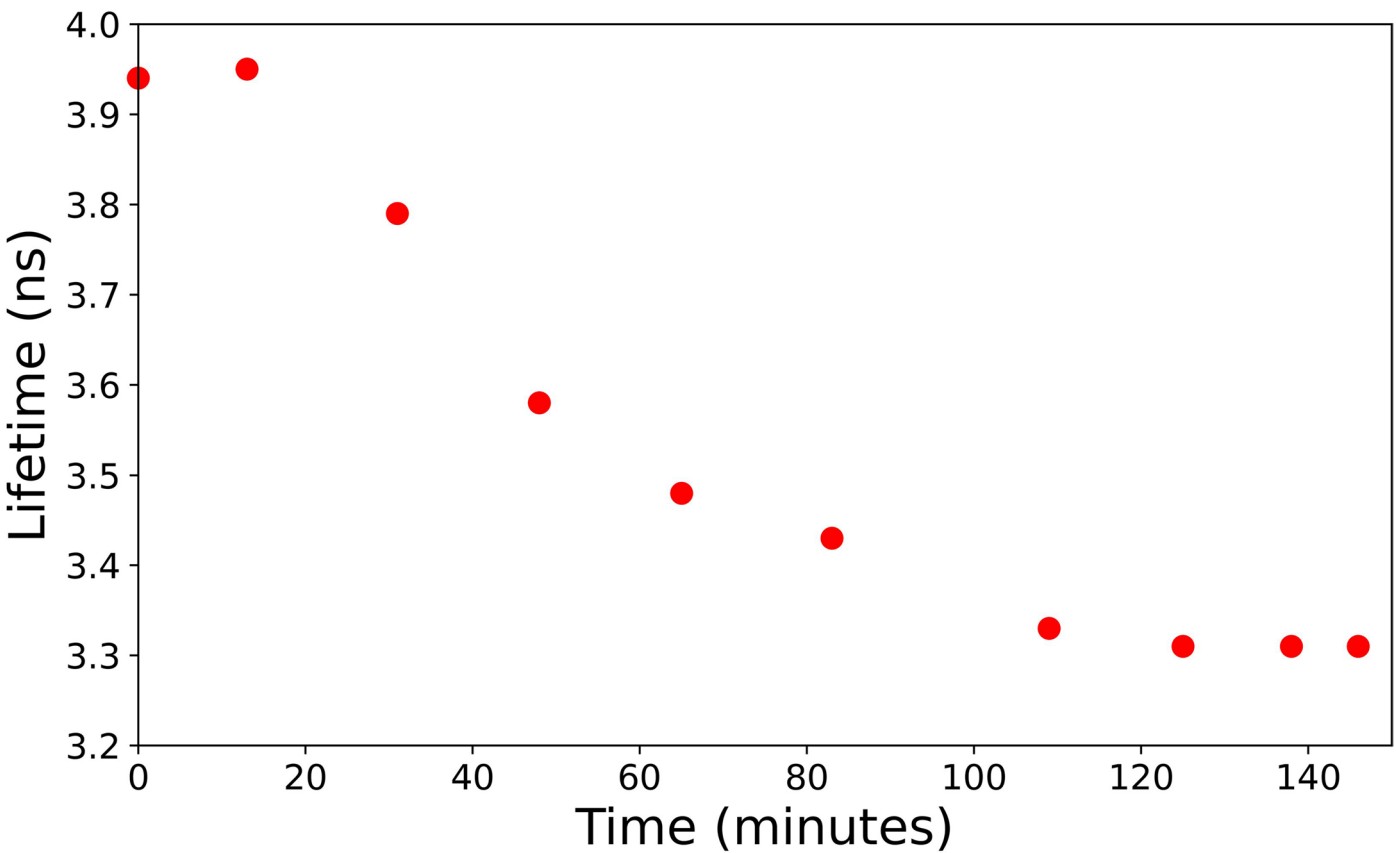

**Extended Data Fig. 4 | Membrane tension of the appressorium decreases during maturation.** Guy11 appressoria were stained with the rotor probe N⁺-BDP at 4.5 hpi (0 min) and FLIM images captured for 3 h and fluorescence lifetimes plotted.

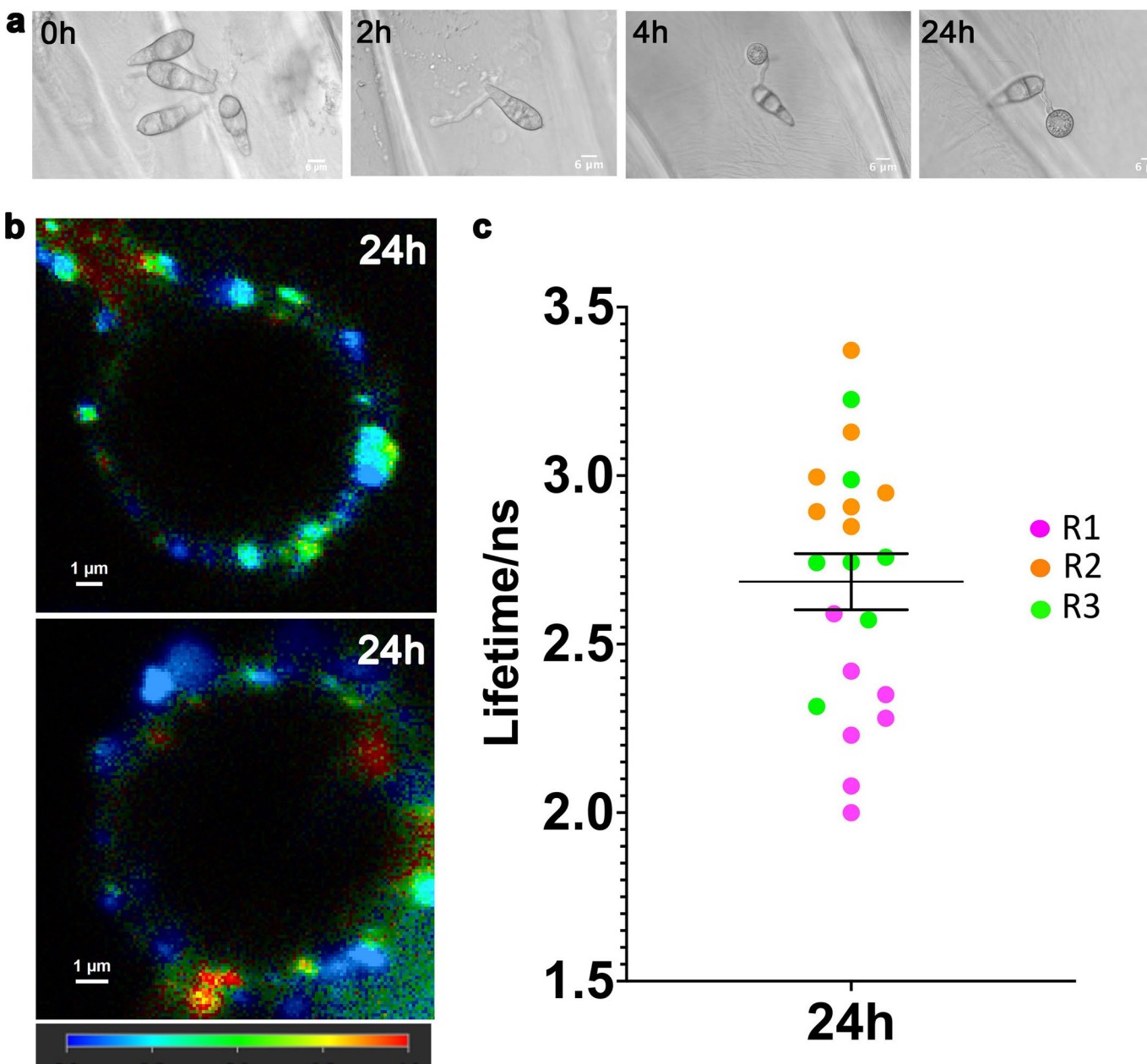

**Extended Data Fig. 5 | The mechanosensor N⁺-BDP reveals spatial variations in membrane tension in *M. oryzae* appressoria on a yielding plant surface. a**, Time course of *M. oryzae* infection-related-development on onion epidermis treated with chloroform. Images are representative of *n = 3* independent repeats of the experiment. **b**, FLIM micrographs of Guy11 at 24 h germinated on onion epidermis treated with chloroform and stained with the rotor probe N⁺-BDP. The colour corresponds to the fluorescence lifetime values expressed in nanoseconds, as shown in the key 2–4 ns. **c**, Dot plot showing the average fluorescence lifetime for 24hpi Guy11 appressoria developed on onion epidermis. Each dot corresponds to the average fluorescence lifetime obtained for an ROI drawn around the membrane of an individual appressorium in a 2D FLIM image. Total observations *n* = 21 appressoria in 3 biological replicates; each biological replicate is colour coded (R1: replicate 1, R2: replicate 2, R3: replicate 3); data are presented as mean ± SEM.

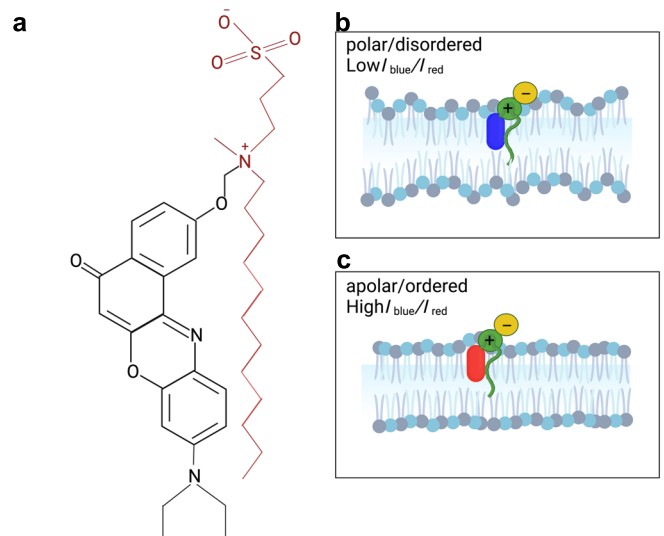

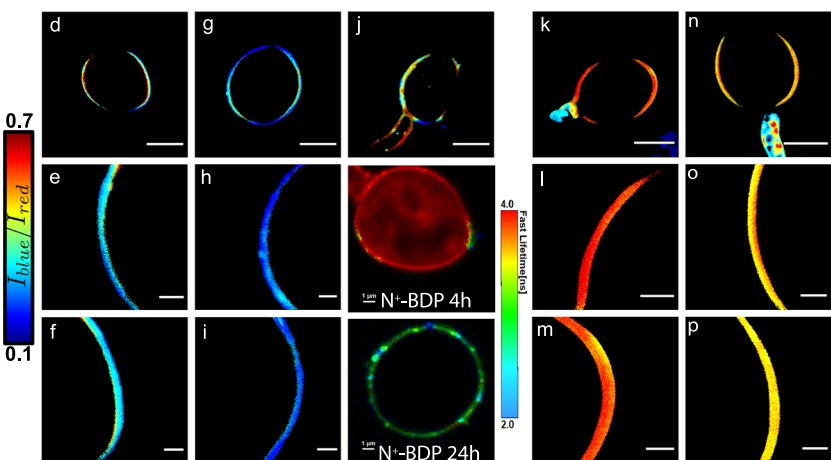

**Extended Data Fig. 6 | Mapping spatial variation in chemical polarity of the plasma membrane in *M. oryzae* appressoria using the solvatochromic probe NR12S. a**, Chemical structure of NR12S. **b,c** Schematic illustrations showing the mechanism by which NR12S reports changes in chemical polarity and lipid phase. **d-j**, Intensity ratio chemical polarity maps in *M. oryzae* wild type strain Guy11 4 h appressoria. Images **e, f, h** and **i** are magnified areas of 4 h appressoria. The colour scale translates the intensity ratio values ($n$ = 31 three independent repetitions of the experiment were performed). N⁺-BDP 4 h and N⁺-BDP 24 h FLIM images included for comparison. **k-p**, Intensity ratio chemical polarity maps of *M. oryzae* wild type strain Guy11 24 h appressoria. Images **l, m, o** and **p** are zoomed in areas of 24 h appressoria, Total observations $n$ = 34 appressoria in 3 biological replicates. Images are representative of $n$ = 3 independent repeats of the experiment. Images **d**, **g**, **j**, **k** and **n**, scale bars = 5 μm, images **e**, **f**, **h**, **i**, **l**, **m**, **o** and **p** scale bars= 1 μm. Image created with BioRender.com.

# Reporting Summary

## Statistics

For all statistical analyses, confirm that the following items are present in the figure legend, table legend, main text, or Methods section.

| n/a | Confirmed | |
|---|---|---|
| ☐ | ☒ | The exact sample size (*n*) for each experimental group/condition, given as a discrete number and unit of measurement |
| ☐ | ☒ | A statement on whether measurements were taken from distinct samples or whether the same sample was measured repeatedly |
| ☐ | ☒ | The statistical test(s) used AND whether they are one- or two-sided *Only common tests should be described solely by name; describe more complex techniques in the Methods section.* |
| ☒ | ☐ | A description of all covariates tested |
| ☒ | ☐ | A description of any assumptions or corrections, such as tests of normality and adjustment for multiple comparisons |
| ☐ | ☒ | A full description of the statistical parameters including central tendency (e.g. means) or other basic estimates (e.g. regression coefficient) AND variation (e.g. standard deviation) or associated estimates of uncertainty (e.g. confidence intervals) |
| ☐ | ☒ | For null hypothesis testing, the test statistic (e.g. *F*, *t*, *r*) with confidence intervals, effect sizes, degrees of freedom and *P* value noted *Give P values as exact values whenever suitable.* |
| ☒ | ☐ | For Bayesian analysis, information on the choice of priors and Markov chain Monte Carlo settings |
| ☒ | ☐ | For hierarchical and complex designs, identification of the appropriate level for tests and full reporting of outcomes |
| ☒ | ☐ | Estimates of effect sizes (e.g. Cohen's *d*, Pearson's *r*), indicating how they were calculated |

*Our web collection on statistics for biologists contains articles on many of the points above.*

## Software and code

Policy information about availability of computer code

| Data collection | A real time FLIM movie was captured for appressorium turgor generation between 4-7.5h over 146 min. FLIM imaging was performed on a Stellaris 8 Falcon upright scanning confocal microscope (Leica systems). Other FLIM imaging was performed on a LEICA TCS SP8X upright scanning confocal microscope coupled to a PicoHarp 300 TCSPC module (PicoQuant GmbH). |
|---|---|
| Data analysis | Stellaris images were processed in LAS X (version 4.2, Leica Microsystems). Movies were generated using Python (Anaconda Software Distribution. Computer software. Vers. 3.8.10). Anaconda.2016. LEICA TCS SP8X images were analysed using SymPhoTime 64 (version 2.4, PicoQuant GmbH). The code used to generate the movies for the FLIM time series can be found at https://github.com/SergioGabrielLopez/movie_script. |

For manuscripts utilizing custom algorithms or software that are central to the research but not yet described in published literature, software must be made available to editors and reviewers. We strongly encourage code deposition in a community repository (e.g. GitHub). See the Nature Portfolio guidelines for submitting code & software for further information.

## Data

Policy information about availability of data

All manuscripts must include a data availability statement. This statement should provide the following information, where applicable:
- Accession codes, unique identifiers, or web links for publicly available datasets
- A description of any restrictions on data availability
- For clinical datasets or third party data, please ensure that the statement adheres to our policy

All data that support the findings of this paper are available from the corresponding author on request.

# Human research participants

Policy information about studies involving human research participants and Sex and Gender in Research.

| | |
|---|---|
| Reporting on sex and gender | Not applicable |
| Population characteristics | Not applicable |
| Recruitment | Not applicable |
| Ethics oversight | Not applicable |

Note that full information on the approval of the study protocol must also be provided in the manuscript.

# Field-specific reporting

Please select the one below that is the best fit for your research. If you are not sure, read the appropriate sections before making your selection.

☒ Life sciences  ☐ Behavioural & social sciences  ☐ Ecological, evolutionary & environmental sciences

For a reference copy of the document with all sections, see nature.com/documents/nr-reporting-summary-flat.pdf

# Life sciences study design

All studies must disclose on these points even when the disclosure is negative.

| | |
|---|---|
| Sample size | Sample sizes were as large as practicable for appressorium FLIM capturing and analysis, based on previous studies where estimates have provided statistically significant findings. In all cases we used three biological replicates for experiments, as that is standard practice for our studies in our field to ensure reproducibility of the presented results (for examples: Ryder *et al.,* 2019 Nature 574, pages 423-427, Michels *et al.,* 2020 PNAS 117(30), pages 18110-18118. |
| Data exclusions | No data were excluded from any part of this study. |
| Replication | All experiments were subject to at least three biological replications unless otherwise stated.Technical replications were also carried out as stated in the text Results were consistent between replication unless otherwise stated. |
| Randomization | Microscopy observation and quantification were from samples selected randomly and quantified independently several times. Randomization in experimental procedures (such as membrane imaging for lifetime quantification) was not necessary because experiments were performed under well controlled conditions or treatments. No animal or humans specimens were used in this study  and randomization is not generally used in this field. |
| Blinding | Blind testing was not routinely carried out in the study as it was not relevant to most of the experiments carried out. Blinding was not necessary because every experiment was quantified three independent times with several technical replicates. |

# Reporting for specific materials, systems and methods

We require information from authors about some types of materials, experimental systems and methods used in many studies. Here, indicate whether each material, system or method listed is relevant to your study. If you are not sure if a list item applies to your research, read the appropriate section before selecting a response.

## Materials & experimental systems

| n/a | Involved in the study |
|-----|----------------------|
| ☒ | ☐ Antibodies |
| ☒ | ☐ Eukaryotic cell lines |
| ☒ | ☐ Palaeontology and archaeology |
| ☒ | ☐ Animals and other organisms |
| ☒ | ☐ Clinical data |
| ☒ | ☐ Dual use research of concern |

## Methods

| n/a | Involved in the study |
|-----|----------------------|
| ☒ | ☐ ChIP-seq |
| ☒ | ☐ Flow cytometry |
| ☒ | ☐ MRI-based neuroimaging |

