## [Peer Review File · Nature Microbiology]

Peer Review Information

Journal: Nature Microbiology

Manuscript Title: A molecular mechanosensor for real-time visualization of appressorium membrane tension in *Magnaporthe oryzae*

Corresponding author name(s): Nicholas J. Talbot

Decision Letter, initial version:

Message 17th October 2022

:
Dear Nick,

Thank you for your patience while your manuscript "Direct measurement of appressorium turgor using a molecular mechanosensor in the rice blast fungus *Magnaporthe oryzae*" was under peer review at Nature Microbiology. It has now been seen by our referees, whose expertise and comments you will find at the end of this email. In the light of their advice, we have decided that we unfortunately cannot offer to publish your manuscript in Nature Microbiology.

From the reports, you will see that while they find your work of some potential interest and clearly feel the approach itself is important and will be of interest to the fungal research community, the referees raise concerns about the advance your findings represent over earlier work (given that the probes have been used in plant cells before), the generalizability of the findings for other fungal or oomycete appressoria, and the level of functional and novel biological insight provided (for example, insight into the functional relevance of the heterogeneous regions of membrane tension). The referees also have some important technical concerns, including whether membrane tension can be used as a proxy for turgor. Unfortunately, these criticisms are sufficiently important as to preclude publication of your work in Nature Microbiology.

Although we regret that we cannot offer to publish your paper in Nature Microbiology, I have discussed your manuscript and the reviewers' comments with our colleagues at Nature Communications, and they are very interested. They would send an appropriately revised version back to the reviewers if the manuscript is transferred to their journal. If you wish to have your revised paper considered by Nature Communications, please use the link to the Springer Nature manuscript transfer service in the footnote once the revision is ready, and include a point-by-point response to the reviewers' concerns. Your handling editor at Nature Communications would be Cesar Sanchez (cesar.sanchez@nature.com). Please feel free to contact him if you wish to discuss the revisions; the most effective way of doing this would be to email him (before starting working on the revisions) a complete, tentative point-by-point response, describing how you plan to address each point. Please note that Nature Communications is a fully open access journal; for information about article processing charges, open access funding, and advice and support from Nature Research, please consult the Nature Communications Open Access page (nature.com/ncomms/about/open-access).

I am very sorry that we cannot be more positive on this occasion, but hope that you find the referees' comments helpful when preparing your paper for resubmission elsewhere.

[redacted]

Reviewer Expertise:

Referee #1: *Magnaporthe oryzae*
Referee #2: Fungal cell biology, mechanobiology
Referee #3: Biomechanics of Plant Development
Referee #4: Fungus-plant interactions

Reviewers Comments:

Reviewer #1 (Remarks to the Author):

Plant pathogenic fungi and oomycetes have similar morphologies and cause significant damage and losses to important crops. Species of both often use appressorial cells to penetrate the outer layer of plant cells. Appressoria can be pressurized through the generation of turgor that acts on a peg at the appressorial base, and the best studied of this type are those of the rice blast fungus *Magnaporthe oryzae*. Studies by this group and others have identified genes, such as those involved in melanin biosynthesis, that are required for turgor and, in one case by this group, the *SLN1* gene required to prevent excess turgor, but the effect of mutations on turgor has not always been easy to measure due to a reliance on the incipient cytorrhysis assay, which indirectly measure appressorium turgor. In this article, the authors provide a “cool tool” proof-of-concept study by showing how a membrane-targeting molecular mechanoprobe can be combined with Fluorescence Lifetime Imaging to quantify changes in membrane tension as proxy for appressorial turgor. As appressoria mature and turgor is generated, membrane tension increases, leading to more rotation of a boron-dipyrromethene (BODIPY)-based molecular rotor. This leads to reduced fluorescence lifetime of the mechanoprobe. Therefore, as the appressorium develops and membrane tension increases with turgor, the authors show the rotor lifetime decreases. Furthermore, the authors show this decrease in rotor lifetime is dependent on melanin biosynthesis, as would be expected if melanin is required for turgor generation. Rotor lifetime is increased (ie tension is decreased) when turgor is artificially lowered using glycerol. Rotor lifetime is decreased (ie tension is increased) below wild type levels in the *sln1* mutant that lacks the turgor sensor, again in line with prediction. Rotor lifetimes are successfully tested in other mutants, and temporally matched to the formation of the septin ring during appressorial development.

In general, this was an interesting study with very high-quality images and movies that support use of the N⁺-BDP mechanosensory in measuring turgor and also in illuminating the large-scale spatial homogeneities in membrane mechanics at high pressure. Thus, this work shows how probing the vast internal pressures of appressoria can be improved over current methods. It also taps into recent developments and excitement around fungal mechanobiology, an emerging field capitalizing on molecular probe breakthroughs. However, I have some concerns and comments that might be addressed before publication.

Main

1. *Magnaporthe* appressoria adhere tightly to hard surfaces. This seals the appressorial pore to the substrate surface, thereby preventing solute leakage and facilitating turgor build-up. Early studies (see Howard and Valent, *Breaking and Entering*, 1996, for example) found how appressorial adhesion could be disrupted when appressoria were formed on permeable soft substrates like cellophane, resulting in loss of turgor. However, measuring loss of turgor under these conditions by the incipient cytorrhysis method is not ideal. Considering some recent work has revisited appressorial adhesion, in order to more comprehensively test the performance of the rotor under all likely usage scenarios, I wonder if the N⁺-BDP probe could be used to measure appressorial turgor on soft surfaces.

2. The studies on appressorial turgor described here were all performed on inductive artificial surfaces. I wonder if the N⁺-BDP probe can be used to measure the turgor of appressoria during development on host rice leaf surfaces? It would be very exciting to assess turgor development up to and beyond the point of host cell penetration. Are these types measurements possible, or are there some limitation to using the probe in planta that should be discussed here?

3. I was uncertain about what the a solvatochromic Nile red-based probe was measuring and why it was included here. I found this section a little confusing, perhaps because unlike Fig 1c, a diagram of the action of this probe was not included. Also, the meaning of the results are confusing. For example, changes in polarity appear to due to a number of factors such as hydration and protein composition that were not resolved, so what is the

point of including this data here? Also, if NR12S displays homogenous polarity or variations that do not match larger changes observed with the N+-BDP rotor, how does this lead to the conclusion that N+-BDP can reveal changes without being affected by polarity? Might it not instead be the case that NR12S is just worse at binding appressorial membranes? Please clarify.

4. The mechanoprobe showed the *sln1* mutant is increased for turgor at 24 h compared to Guy11 as expected, but it also showed increased *sln1* appressoria turgor at 4 h (fig 4e). Why would this be the case? My understanding is that the *Sln1* turgor sensor triggers invasion when a sufficient turgor threshold has been crossed, but does the affect at 4 h mean that *Sln1* is monitoring turgor throughout appressorial development? Can the authors expand on this? Thus, the *Sln1* data needs to be discussed more. Also, with regards to this section of the paper, the authors note that the probe might become saturated at 8 h as the average lifetime of the rotor did not significantly change after 8 h. If so, then isn't probe saturation at 8 h a big problem for future studies using this probe? How would such saturation affect its utility and accuracy going forward? Can saturation be remediated?

5. The results are convincing for *Magnaporthe* appressoria, but can this probe work well with other appressoria, for example those of *Colletotrichum* or a species of oomycete? It might be necessary to show that this probe can be widely deployed and is not limited to *Magnaporthe* due to, for example, the membrane composition of that particular species.

Minor

Supplementary Video 2- the text suggests this video was made at 24 h but the video title says 7.5 h. Also, the video links are not labeled with numbers, making it hard to know which ones are which.

Extended Data 1. Is there an explanation for why fluorescence lifetime decreased first at 0.4 M and 0.75 M glycerol before increasing again at 1M?

Comparing Fig 2d to 2e: The rim of appressoria in Fig 2d is green, whereas those of Figs 2a and 2b are red, yet fig 2e shows they have similar lifetimes. I am confused on this point.

Supplementary Video 5 (which is not labelled as such) is out of order in the uploaded files.

Reviewer #2 (Remarks to the Author):

Plant diseases caused by pathogenic fungi and oomycetes have significant impacts on biodiversity and ecosystems, as well as contributing to crop losses in agriculture, horticulture and forestry. These impacts are likely to increase in the future due to the impact of climate change. It is therefore of utmost importance to understand the mechanisms that these organisms use to infect their hosts. One aspect that has seen less study, due largely to technical difficulties, is the biophysical means of "forcing" entry into the host. Some pathogens such as the rice blast fungus *Magnaporthe oryzae* develop huge internal hydrostatic pressures in specialized infection structures called appressoria, which enable penetration of the host. The incorporation of melanin in the cell wall is crucial for the development of such pressure as it prevents the loss of solutes from the appressoria. The magnitude of these pressures and/or the small size of cells makes their study difficult using conventional methodology.

In the study of Rydel et al. a relatively new technique of using the molecular membrane mechanoprobe N+BDP is used. This probe is responsive to tension in the membrane, which can be used as an indication of pressure. It has been used on the wild type strain

Guy11 and various mutants deficient in melanin synthesis, septin (a crucial component of appressoria development) synthesis and a kinase responsible for turgor sensing. The study effectively shows that the probe can be used to measure tension in the membrane and intriguingly shows a heterogeneous distribution of tension in the appressoria membrane. This is certainly of great interest to the fungal/plant pathology community and the technique promises much in the future.

Unfortunately, at present I am unconvinced however that the data presented would warrant publication in Nature Microbiology, rather than in a top fungal/cell/plant pathology journal. Given that these probes have already been used in plant cells, this is not the first time that their use has been documented. This therefore compromises the novelty of their use. This could have been overlooked had the authors been able to show a functional aspect of these microdomains of high tension. An example of this is their suggestion in the Discussion that they could be associated with points of penetration. So, if they were able to show something akin to this then publication would have been warranted.

Overall, the standard of science and writing is good. I have made a few suggestions that the authors may consider, in addition to any further experimental results, for any resubmission.

- Throughout the manuscript: There appears to be some inconsistency relating to what the authors say the probe is measuring. Thus, it is described as "directly measuring turgor" (in the paper title), "a direct proxy for turgor", something that "directly visualizes turgor". I would suggest that none of these are technically correct, as in indicating membrane tension the probe is measuring something that is a consequence of turgor pressure. Some more consistency here is needed – the use of "direct" is perhaps a little strong.
- Line 58. Could a sentence be added to indicate how high this is in comparison to vegetative hyphae (e.g. cite some of the pressure probe studies on fungi/oomycetes) to really emphasize the pressures involved?
- On line 194 – it was a little unclear exactly what was meant by increasing hydration of the membrane.
- It is a shame that only one septin mutant (Δ sep5) was tested. Why was sep5 chosen?
- Paragraph starting line 261: I found the kinase mutation results of more interest than the septin mutations. Could this data be presented earlier?
- Line 267: The term "excess, runaway turgor" is a little too conversational.
- Line 316: Appressoria in oomycetes (effectively apical swellings) are somewhat distinct to those in fungi and so this should be noted. Possibly by describing them as "functional appressoria"?
- Line 320: Insert a space between 8 and MPa. Also see comment re line 58 – same applies here.
- Line 373: A mathematical model was mentioned, but there appeared to be no reference to this.
- Line 345: Presumably there would be some turgor generation to enable a turgid cell - it would just not be at the same levels of magnitude as that in the appressoria.
- Figure 1/Extended Fig 3: As these indicate the heterogeneous nature of the mechano-probe versus the more homogeneous nature of the chemical probe, it would help if parts of these could be combined to enable easier comparison. This is described well in the text, but to avoid flipping between Figs putting one image of the chemical probe in Fig 1, or one

image of the mechano-probe into extended Fig 3 would be helpful.

- Extended Fig 4: It appears that part of the Fig on the left has been blacked out?

Reviewer #3 (Remarks to the Author):

In the present manuscript a fluorescent reporter for membrane tension is used to assess the turgor status of conidia of the rice blast fungus. This new technical approach has been long awaited since turgor status is a crucial parameter not only for the pathogenic action of fungi, but for many other cell and tissue systems including those of plants. This manuscript is therefore of fundamental importance for the field.

Despite the excitement, I have a few fundamental questions with regards to the use of the fluorescent reporter (points 1 & 2) and a technical question (point 3):

1. I am not entirely convinced of the concept that membrane tension can be used as a proxy for turgor. While the correlation between the FLIM signal and the expected turgor status of the cells seems to be compelling, I am wondering why the plasma membrane in walled cells would actually be expected to be under tension when the cell is under higher turgor? The hydrostatic pressure is actually withstood by the wall (outer stiff layer), not by the membrane (inner soft layer). While an instantaneous (or rapid) change of turgor can indeed be expected to also put the plasma membrane under tension since on a short time scale the amount of lipids in the membrane surface remains constant while the cell volume (and thus the magnitude of the surface) increases slightly, at longer time scales (e.g. those relevant for the development of an appressorium) one could expect the cell to add lipids to the plasma membrane surface through exocytosis, thus re-establishing the amount of lipid per unit surface. In fact, the internal soft layer of a pressurized vessel does not need to be under in-plane tension at all for the pressure to put the external stiff layer under tension. How does this concept play into the interpretation of the data? This conceptual concern with membrane tension as a proxy for turgor might have been addressed in some of the cited papers (which I did not have the time to consult), and it would be important to spell out the answers to these concerns clearly.

2. The FLIM images show clearly that the membrane tension seems to be quite heterogeneous over the cell surface. This is difficult to reconcile with the fact that in a spherical pressurized body, the tensile stress in the loadbearing wall is uniform across the surface (assuming uniform thickness and mechanical properties of the wall). So how are these huge local differences (e.g. Fig. 3) explained?

3. Point 2 leads me to a technical question: How was the average fluorescence lifetime calculated on these heterogeneously labeled images? Which pixels on the spherical conidium were used for the average? Please provide a very detailed description of the method that allows reproducing the experiments for the non-expert.

Reviewer #4 (Remarks to the Author):

In their manuscript, the authors report on the development and use of a molecular mechanosensor to visualise membrane tension as a measure for turgor in the rice blast fungus *Magnaporthe oryzae*. This fungus has been reported to produce appressoria that are able to build an enormous turgor pressure, which makes it an attractive model to test these mechanosensors in. By making use of diverse mutants that are known to be affected in appressorial development or functioning, the authors convincingly show that the mechanosensor differentially response to diverse turgor pressures, and can be used as a read-out for such pressure. Interestingly, their study shows that membranes are perhaps

not as homogenous as previously thought, and that spatial heterogeneity occurs.

Overall, the manuscript is very clear, well written, and convincing. The data that are produced are robust, and the conclusions justified based on the data that are presented. Admittedly, I read this paper mostly as a technical advance, as I feel that the advance in our understanding of the (infection)biology of *Magnaporthe* is limited, and there where such advance is provided (e.g. the heterogeneous nature of the appressorial membrane) the advance remains quite descriptive.

In their title, the authors refer to "Direct *measurement* of appressorium turgor", but after reading the manuscript I am not so sure whether this is indeed what the authors did. That is, I see mention of more and of less membrane tension, but this is not extrapolated to turgor measurements, leading to infernal of turgor pressure values. Can the authors comment on whether inferring such values is possible based on the currently presented technology? And if not, what would be required to be able to do so?

Also, I am wondering to what extent their observations are a direct consequence of membrane tensions versus degrees of melanin deposition that somehow affect the mechanosensors? I think in most of the mutants this is all correlated; less melanin, less turgor, but can the authors somehow uncouple these traits?

[redacted]

Author Rebuttal to Initial comments

Please find below our responses to the comments of the reviewers following the first submission of the review process for *Nature Microbiology*. We have addressed the reviewer comments below in red, and would be very grateful if you would consider our revised manuscript for publication in *Nature Microbiology*.

Reviewer #1 (Remarks to the Author):

Plant pathogenic fungi and oomycetes have similar morphologies and cause significant damage and losses to important crops. Species of both often use appressorial cells to penetrate the outer layer of plant cells. Appressoria can be pressurized through the generation of turgor that acts on a peg at the appressorial base, and the best studied of this type are those of the rice blast fungus *Magnaporthe oryzae*. Studies by this group and others have identified genes, such as those involved in melanin biosynthesis, that are required for turgor and, in one case by this group, the *SLN1* gene required to prevent excess turgor, but the effect of mutations on turgor has not always been easy to measure due to a reliance on the incipient cytorrhysis assay, which indirectly measure appressorium turgor. In this article, the authors provide a "cool tool" proof-of-concept study by showing how a

membrane-targeting molecular mechanoprobe can be combined with Fluorescence Lifetime Imaging to quantify changes in membrane tension as proxy for appressorial turgor. As appressoria mature and turgor is generated, membrane tension increases, leading to more rotation of a boron-dipyrromethene (BODIPY)-based molecular rotor. This leads to reduced fluorescence lifetime of the mechanoprobe. Therefore, as the appressorium develops and membrane tension increases with turgor, the authors show the rotor lifetime decreases. Furthermore, the authors show this decrease in rotor lifetime is dependent on melanin biosynthesis, as would be expected if melanin is required for turgor generation. Rotor lifetime is increased (ie tension is decreased) when turgor is artificially lowered using glycerol. Rotor lifetime is decreased (ie tension is increased) below wild type levels in the *sln1* mutant that lacks the turgor sensor, again in line with prediction. Rotor lifetimes are successfully tested in other mutants, and temporally matched to the formation of the septin ring during appressorial development.

In general, this was an interesting study with very high-quality images and movies that support use of the N⁺-BDP mechanosensory in measuring turgor and also in illuminating the large-scale spatial homogeneities in membrane mechanics at high pressure. Thus, this work shows how probing the vast internal pressures of appressoria can be improved over current methods. It also taps into recent developments and excitement around fungal mechanobiology, an emerging field

capitalizing on molecular probe breakthroughs. However, I have some concerns and comments that might be addressed before publication.

Main

1. Magnaporthe appressoria adhere tightly to hard surfaces. This seals the appressorial pore to the substrate surface, thereby preventing solute leakage and facilitating turgor build-up. Early studies (see Howard and Valent, Breaking and Entering, 1996, for example) found how appressorial adhesion could be disrupted when appressoria were formed on permeable soft substrates like cellophane, resulting in loss of turgor. However, measuring loss of turgor under these conditions by the incipient cytorrhysis method is not ideal. Considering some recent work has revisited appressorial adhesion, in order to more comprehensively test the performance of the rotor under all likely usage scenarios, I wonder if the N⁺-BDP probe could be used to measure appressorial turgor on soft surfaces.

We are very grateful to the reviewer for this comment. In our experience, it is very difficult to find 'soft surfaces' that will reproducibly induce appressorium formation. We have found that cellophane induction, for example, still requires mounting on a harder surface for appressorium induction. To address this comment, and the helpful point made about plant surfaces, we decided to carry out appressorium development on sterile onion epidermis (shown in Extended Data Fig. 4a-c). We applied N⁺-BDP and found that 24h appressoria share a similar spatial membrane heterogeneity to that observed in appressoria developed on artificial coverslips and had a similar average lifetime of 2.69 ± 0.083 ns. This suggests that turgor still accumulates on a yielding plant surface.

2. The studies on appressorial turgor described here were all performed on inductive artificial surfaces. I wonder if the N⁺-BDP probe can be used to measure the turgor of appressoria during development on host rice leaf surfaces? It would be very exciting to assess turgor development up to and beyond the point of host cell penetration. Are these types measurements possible, or are there some limitation to using the probe in planta that should be discussed here?

We agree with the reviewer and for this reason carried out the onion epidermis infection assays. We would like to use rice epidermis, but the N⁺-BDP reacts with all membranes that it encounters. This makes it difficult to separate the fungal plasma membrane signal in the appressorium from the plant plasma membrane in the epidermal cells being penetrated. To overcome this problem, we used onion epidermis, which was first sterilised using CH₃Cl treatment. This prevents N⁺-BDP reacting with the epidermal membranes, so that it is specific to fungal structures. In this way, we were able to visualise appressorium development on a yielding plant surface.

Future studies in which a FLIM probe is designed to exclusively target the fungal plasma membrane would allow for rice penetration experiments to be performed.

These studies are planned for the future, once a N⁺-BDP derivative with such specificity is developed. This is, however, beyond the scope of the current submission.

3. I was uncertain about what the solvatochromic Nile red-based probe was measuring and why it was included here. I found this section a little confusing, perhaps because unlike Fig 1c, a diagram of the action of this probe was not included.

We thank the reviewer for this comment and suggestion. The explanation for the purpose of the solvatochromic Nile-red-based probe is provided below and in the revised manuscript. We have now included a cartoon in Extended Data Fig.5a to summarise the action of this probe, as helpfully suggested by the reviewer.

Also, the meaning of the results are confusing. For example, changes in polarity appear to due to a number of factors such as hydration and protein composition that were not resolved, so what is the point of including this data here? Also, if NR12S displays homogenous polarity or variations that do not match larger changes observed with the N⁺-BDP rotor, how does this lead to the conclusion that N⁺-BDP can reveal changes without being affected by polarity? Might it not instead be the case that NR12S is just worse at binding appressorial membranes? Please clarify.

We thank the reviewer for these thoughtful comments. The N⁺-BDP probe is primarily sensitive to changes in membrane tension, but it has been reported to be somewhat solvatochromic– sensitive to changes in lipid composition in membranes. Therefore, we needed to carry out a *control* experiment to test whether the observed heterogeneity in fluorescence lifetime in pressurized appressoria was due to tension, or due instead to the formation of lipid microdomains. The NR12S probe was used for this purpose. It is a probe which is insensitive to tension and only sensitive to compositional inhomogeneities. Staining appressoria, at the same time points (4 h and 24 h), with this probe revealed that there were no substantial inhomogeneities in lipid composition. There are no distinct chemical microdomains in the appressorium membranes, and the observed heterogeneities in lifetime with N⁺-BDP can therefore only be attributed to a very inhomogeneous tension across the membrane. Both probes have an excellent capacity to bind appressorium membranes, as seen from intensity-based images utilizing a similar chemical strategy– the attachment of a lipid tail to the fluorescent reporter unit. To avoid confusion, we have now re-worded our description in the manuscript of the NR12S probe to make these points clear.

4. The mechanoprobe showed the *sln1* mutant is increased for turgor at 24 h compared to Guy11 as expected, but it also showed increased *sln1* appressoria turgor at 4 h (fig 4e). Why would this be the case? My understanding is that the Sln1 turgor sensor triggers invasion when a sufficient turgor threshold has been crossed, but does the affect at 4 h mean that Sln1 is monitoring turgor throughout appressorial development? Can the authors expand on this? Thus, the Sln1 data needs to be discussed more. Also, with regards to this section of the paper, the authors note that the probe might become saturated at 8 h as the average lifetime of the rotor did not significantly change after 8 h. If so, then isn't probe saturation at 8 h a big problem for future studies using this probe? How would such saturation affect its utility and accuracy going forward? Can saturation be remediated?

We thank the reviewer for this comment and we have now expanded discussion of this observation in the manuscript, as suggested. We believe our analysis of the *!!sln1* mutant actually demonstrates the power of the N+-BDP FLIM analysis. As the reviewer

states, this technique shows that the *!sln1* mutant does exhibit enhanced turgor pressure throughout a time course of infection-related-development. Our original model for the role of the Sln1 sensor kinase was that it negatively regulates turgor once a sufficient threshold of pressure has been reached (Ryder et al., 2019 *Nature*), and that this facilitates subsequent repolarisation of the appressorium and plant infection. Our previous studies of turgor within the $\Delta sln1$ mutant were, however, all carried out using the incipient cytorrhysis assay. The N+-BDP FLIM analysis suggests that Sln1 in fact modulates turgor throughout inflation of the appressorium, suggesting that the regulated build-up of turgor in the infection cell requires Sln1. Without this, there is excess turgor even at early timepoints.

The "saturation" of the probe response is an intrinsic property of molecular rotor-based rigidochromic dyes, because the response is bounded by a state in which rotation can occur freely (under high tension conditions) on the one side, and a state in which rotation is fully blocked on the other side. Since this is an intrinsic physical feature of the operational mechanism of these probes this cannot be remediated. However, while we acknowledge that this is a limitation of their use, hence our mention of this effect in the manuscript, it is still the only available tool to probe turgor development in live-cell imaging in a non-invasive way, and thereby still allows us to study the mechanical dimension of host entry in a way that was in-accessible before. This is particularly so, when considering the huge turgor that we are monitoring and the enormous invasive force thereby generated. Monitoring such pressures in living cells has not been previously possible.

5. The results are convincing for Magnaporthe appressoria, but can this probe work well with other appressoria, for example those of *Colletotrichum* or a species of oomycete? It might be necessary to show that this probe can be widely deployed and is not limited to Magnaporthe due to, for example, the membrane composition of that particular species.

We are grateful to the reviewer for this suggestion. We have now imaged early and late appressoria in the hemibiotrophic ascomycete fungus *Colletotrichum higginsianum* and shown that early stage appressoria display a homogenous higher fluorescent lifetime when compared to later appressoria, which display a lower average fluorescent lifetime, consistent with increased tension. These results corroborate our findings in *M. oryzae* and they confirm the mechanoprobe can be deployed in other appressorium-forming fungi.

Minor

Supplementary Video 2- the text suggests this video was made at 24 h but the video title says 7.5 h. Also, the video links are not labeled with numbers, making it hard to know which ones are which.

We thank the reviewer for this observation and have amended the labelling accordingly.

Extended Data 1. Is there an explanation for why fluorescence lifetime decreased first at 0.4 M and 0.75 M glycerol before increasing again at 1M?

The change in lifetime between 0.4M and 0.75M is not statistically significant when compared to the control. The fact that the curve deviates at low osmolarity could be due to various factors: i) it could be that the minimum in the curve corresponds to the isotonic condition for the pathogen at that time point? (we know that the membrane response is not symmetric to tension vs compression). ii) membranes undergo large phase changes as a function of membrane tension. The probe only has a predictable response when only the tension changes and nothing else. The minimum in the lifetime vs osmolarity could signal a tension-induced transition in the membrane. There is quite a lot of biophysical literature about that fact that tension can make homogeneous membranes phase separate into islands or change the structure and topology of membrane microdomains. We make reference to these factors, but we cannot comment on an apparent 'difference' which is not statistically significant.

Comparing Fig 2d to 2e: The rim of appressoria in Fig 2d is green, whereas those of Figs 2a and 2b are red, yet fig 2e shows they have similar lifetimes. I am confused on this point.

We thank the reviewer for this comment. The colour is different between panels due to the cut-off point on the lifetime scale. To highlight the lack of spatial membrane heterogeneity in the tricyclazole-treated control, we have included black and white images alongside the colour images in the revised figures (now Fig. 3). Treatment with the melanin biosynthesis inhibitor tricyclazole will not perfectly replicate the phenotype of the melanin-deficient mutants, however, it is sufficient to disrupt melanin production and prevent rice blast disease.

Supplementary Video 5 (which is not labelled as such) is out of order in the uploaded files.

We thank the reviewer for this observation and have corrected the upload order.

Reviewer #2 (Remarks to the Author):

Plant diseases caused by pathogenic fungi and oomycetes have significant impacts on biodiversity and ecosystems, as well as contributing to crop losses in agriculture, horticulture and forestry. These impacts are likely to increase in the future due to the impact of climate change. It is therefore of utmost importance to understand the mechanisms that these organisms use to infect their hosts. One aspect that has seen less study, due largely to technical difficulties, is the biophysical means of "forcing" entry into the host. Some pathogens such as the rice blast fungus *Magnaporthe oryzae* develop huge internal hydrostatic pressures in specialized infection structures called appressoria, which enable penetration of the host. The incorporation of melanin in the cell wall is crucial for the development of such pressure as it prevents the loss of solutes from the appressoria. The magnitude of these pressures and/or the small size of cells makes their study difficult using

conventional methodology.

In the study of Rydel et al. a relatively new technique of using the molecular membrane mechanoprobe N+BDP is used. This probe is responsive to tension in the

membrane, which can be used as an indication of pressure. It has been used on the wild type strain Guy11 and various mutants deficient in melanin synthesis, septin (a crucial component of appressoria development) synthesis and a kinase responsible for turgor sensing. The study effectively shows that the probe can be used to measure tension in the membrane and intriguingly shows a heterogeneous distribution of tension in the appressoria membrane. This is certainly of great interest to the fungal/plant pathology community and the technique promises much in the future.

Unfortunately, at present I am unconvinced however that the data presented would warrant publication in Nature Microbiology, rather than in a top fungal/cell/plant pathology journal. Given that these probes have already been used in plant cells, this is not the first time that their use has been documented.

We believe that the observations made in this study are novel and original. This is the first study in which a mechanosensor probe has been used to study turgor-mediated plant infection in a pathogenic fungus. This pathogen develops pressure, which is an order of magnitude higher than any pressure observed in plant or animal cells. We would argue that the novelty of the study is showing that such enormous pressure (8.0 MPa or 80 atmospheres) can be directly visualised and evaluated in living cells. Furthermore, we show that membranes under such extreme pressures show heterogeneity in tension, which is a phenomenon that has not previously been observed in living cells.

This therefore compromises the novelty of their use. This could have been overlooked had the authors been able to show a functional aspect of these microdomains of high tension. An example of this is their suggestion in the Discussion that they could be associated with points of penetration. So, if they were able to show something akin to this then publication would have been warranted.

We would stress that the large heterogeneities in membranes placed under such large amounts of tension by enormous turgor pressures, have not been previously observed. This is a novel finding in and of itself. While there have been ample studies in unwallled cells, or free-standing *in vitro* membranes, that link tension to changes in membrane homogeneity (Garcia-Sáez et al., 2007 J. Biol Chem. 282, 33537-33544; Sens and Turner, 2006 Physical Rev E 73, 031918), these cannot be pressurized to the same extent as membranes enclosed in appressoria. The findings presented here are therefore the very first report of membrane heterogeneity under enormous pressures, and these results highlight how these huge levels of tension give rise to regimes in membrane physics that were unknown to date.

The second area of real novelty in the report is the demonstration that the Sln1 turgor sensor is monitoring turgor throughout appressorium inflation and maturation. We previously demonstrated that the Sln1 histidine-aspartate kinase acts as a turgor sensor in the appressorium. Our original model, however, was that the kinase would modulate turgor once a threshold level had been reached. Based on our

nature portfolio

mechanosensory analysis here it seems much more likely that Sln1 is necessary for the regulated generation of turgor and that removing the kinase leads to excess throughout appressorium development. This highlights that appressorium turgor

generation is a tightly regulated process in which the rate of turgor increase is important.

Overall, the standard of science and writing is good. I have made a few suggestions that the authors may consider, in addition to any further experimental results, for any resubmission.

- Throughout the manuscript: There appears to be some inconsistency relating to what the authors say the probe is measuring. Thus, it is described as “directly measuring turgor” (in the paper title), “a direct proxy for turgor”, something that “directly visualizes turgor”. I would suggest that none of these are technically correct, as in indicating membrane tension the probe is measuring something that is a consequence of turgor pressure. Some more consistency here is needed – the use of “direct” is perhaps a little strong.

We thank the reviewer for this comment. We have tried to remove inconsistency in terminology throughout the manuscript. We have adopted the term ‘visualisation of turgor’ predominantly. We also explain this terminology in terms that are consistent with the reviewer statement.

- Line 58. Could a sentence be added to indicate how high this is in comparison to vegetative hyphae (e.g. cite some of the pressure probe studies on fungi/oomycetes) to really emphasize the pressures involved?

We thank the reviewer for this suggestion and have cited some of the pressure probe studies as indicated by the reviewer. We also show non-melanised structures, such as germ tubes in Extended Data 1.

- On line 194 – it was a little unclear exactly what was meant by increasing hydration of the membrane.

We thank the reviewer for this comment and appreciate our phrasing has caused some confusion. Essentially, our intended meaning is that increased tension on the membrane increases the spacing of lipid packing in the bilayer, which places the reporter probe in closer proximity to the water that flanks the bilayer on both sides. This causes it to shift its emission spectrum as the average chemical polarity of the probe’s surroundings is increased. We have modified the description in the manuscript to make this point more clear.

- It is a shame that only one septin mutant (!1sep5) was tested. Why was sep5 chosen?

The $\Delta sep5$ mutant is a core septin in *M. oryzae* and has one of the strongest phenotypes out of the five originally characterised (Dagdaz *et al.*, 2012 *Science*). We do not feel that analysing the rest of the septin mutants is really necessary to make the point and we focused on utilizing the method in another appressorium-

forming fungus in the revision. This experiment can be carried out but we felt it would not provide additional new insight.

- Paragraph starting line 261: I found the kinase mutation results of more interest that

the septin mutations. Could this data be presented earlier?

We are grateful for this comment and we agree with the reviewer. It was actually for this reason that the Sln1 analysis was the final figure in the paper. We felt that this was the most logical way of presenting our study, because the new insight provided the exact role of Sln1 and is very important in our view and also highlights the utility of the new procedure.

- Line 267: The term “excess, runaway turgor” is a little too conversational.

We have removed the term ‘runaway’ which was a term used previously in Ryder et al., 2019, based on editorial suggestions from the *Nature* editors to provide a simple way of indicating the consequence of excess turgor.

- Line 316: Appressoria in oomycetes (effectively apical swellings) are somewhat distinct to those in fungi and so this should be noted. Possibly by describing them as “functional appressoria”?

We are grateful for the comment and have amended the text as suggested by the reviewer.

- Line 320: Insert a space between 8 and MPa. Also see comment re line 58 – same applies here.

We have amended the text, as suggested by the reviewer.

- Line 373: A mathematical model was mentioned, but there appeared to be no reference to this.

We have referenced Ryder et al., 2019 *Nature* as this publication contains the mathematical model of appressorium turgor generation and the prediction that a turgor sensor would be necessary for the cell to operate.

- Line 345: Presumably there would be some turgor generation to enable a turgid cell - it would just not be at the same levels of magnitude as that in the appressoria.

We thank the reviewer for their comment and have amended the text accordingly.

- Figure 1/Extended Fig 3: As these indicate the heterogenous nature of the mechano-probe versus the more homogeneous nature of the chemical probe, it would help if parts of these could be combined to enable easier comparison. This is described well in the text, but to avoid flipping between Figs putting one image of the chemical probe in Fig 1, or one image of the mechano-probe into extended Fig 3 would be helpful.

We thank the reviewer for this suggestion and we have now included a 4 h and 24 h wild type appressorium stained with the rotor probe for ease of comparison between the Figure 1 and Extended Fig. 1.

Extended Fig 4: It appears that part of the Fig on the left has been blacked out?

We thank the reviewer for this observation. The image has not been blacked out, but we aimed to focus on only germ tubes and appressoria for FLIM, so we have cropped the images to reduce the area of displayed black around the germlings.

Reviewer #3 (Remarks to the Author):

In the present manuscript a fluorescent reporter for membrane tension is used to assess the turgor status of conidia of the rice blast fungus. This new technical approach has been long awaited since turgor status is a crucial parameter not only for the pathogenic action of fungi, but for many other cell and tissue systems including those of plants. This manuscript is therefore of fundamental importance for the field.

Despite the excitement, I have a few fundamental questions with regards to the use of the fluorescent reporter (points 1 & 2) and a technical question (point 3):

1. I am not entirely convinced of the concept that membrane tension can be used as a proxy for turgor. While the correlation between the FLIM signal and the expected turgor status of the cells seems to be compelling, I am wondering why the plasma membrane in walled cells would actually be expected to be under tension when the cell is under higher turgor? The hydrostatic pressure is actually withstood by the wall (outer stiff layer), not by the membrane (inner soft layer). While an instantaneous (or rapid) change of turgor can indeed be expected to also put the plasma membrane under tension since on a short time scale the amount of lipids in the membrane surface remains constant while the cell volume (and thus the magnitude of the surface) increases slightly, at longer time scales (e.g. those relevant for the development of an appressorium) one could expect the cell to add lipids to the plasma membrane surface through exocytosis, thus re-establishing the amount of lipid per unit surface. In fact, the internal soft layer of a pressurized vessel does not need to be under in-plane tension at all for the pressure to put the external stiff layer under tension. How does this concept play into the interpretation of the data? This conceptual concern with membrane tension as a proxy for turgor might have been addressed in some of the cited papers (which I did not have the time to consult), and it would be important to spell out the answers to these concerns clearly.

We thank the reviewer for the comment. To generate turgor pressure, an osmotic pressure differential must be generated between the inside and outside of the cell. This requires a semi-permeable barrier, that passes water but not osmolytes. It is the cell membrane that acts as a semi-permeable barrier. Without an intact cell membrane, cells would not be capable of generating turgor, because the osmolytes (typically small sugars or glycols) would rapidly diffuse through the porous cell wall network to equalize pressures. The amount of turgor inside walled cells is so high that

the tension placed on the membrane cannot be resolved by adding additional lipids, since the area of the lipid membrane is bounded by the cell wall that encases it. Even if lipid addition would alleviate the tension caused by an increase partially, the scenario

under study is not a state of mechanostasis but one in which turgor is continuously and rapidly increased over time, so the time scales of lipid biosynthesis and exocytosis versus turgor build-up come into play here. If one would remove the cell wall (e.g. by enzymatic digestion in a protoplast experiment) the cell would inflate to many times its original size and eventually burst, under the conditions of these experiments, indicating how adding lipids to the membrane whose area is bounded by the cell wall can never alleviate the effects of these enormous turgor pressures. The cell wall bounds this inflation by offering an elastic counterforce, and thus bounds the maximum area expansion of the membrane. Thus, in turgescient cells, both the membrane and the cell wall are under tension. There is also ample biological evidence for this, most strikingly the fact that that mechanosensitive ion channels (MSLs and PIEZOs), which are placed in the plasma membrane, are sensitive to turgor generation. If tension would only emerge in the cell wall and that in the membrane is fully alleviated by exocytosis, these ms-ion channels would not be activated in a sustained manner, which is not the case based on the numerous studies of ms-ion channels in plants (for which the argumentation above equally applied).

2. The FLIM images show clearly that the membrane tension seems to be quite heterogeneous over the cell surface. This is difficult to reconcile with the fact that in a spherical pressurized body, the tensile stress in the loadbearing wall is uniform across the surface (assuming uniform thickness and mechanical properties of the wall). So how are these huge local differences (e.g. Fig. 3) explained?

We thank the reviewer for the comment. It is true that mechanics states that an isotropic pressure acting on a perfectly isotropic spherical elastic shell would give rise to a homogeneous tension. However, membranes are not homogeneous isotropic spherical shells, but feature a complex organisation with nano- and microdomains. Moreover, and more likely to be the reason for the observed heterogeneity, is that the membrane is not a free-floating spherical shell, but one that is physically and mechanically anchored to the cell wall through a large variety of proteins that feature a transmembrane and/or extracellular domain. Inhomogeneities in protein anchoring are mostly likely the result of inhomogeneous tension. We note that this is the first observation of these tension inhomogeneities, and that their origin and function, are currently speculative and need future work to be understood. We have addressed these points in the discussion.

3. Point 2 leads me to a technical question: How was the average fluorescence lifetime calculated on these heterogeneously labeled images? Which pixels on the spherical conidium were used for the average? Please provide a very detailed description of the method that allows reproducing the experiments for the non- expert.

For each image we created an ROI by selecting the pixels corresponding to the membrane of the appressorium. The arrival times with respect to the laser pulse of all the fluorescence photons contained within this ROI were used to create a histogram

of arrival times. The histogram was fit using a three-exponential function and the fit was only deemed acceptable if the residuals of the fit were evenly distributed around zero and the Chi2 value was within the 0.70-1.30 range. Then, the average fluorescence lifetime for the image was obtained by calculating the amplitude average

lifetime from the fit parameters. Each repeat consisted of multiple images processed in this manner and the representative average fluorescent lifetime for every time point was calculated from data obtained from three independent repeats.

Reviewer #4 (Remarks to the Author):

In their manuscript, the authors report on the development and use of a molecular mechanosensor to visualise membrane tension as a measure for turgor in the rice blast fungus *Magnaporthe oryzae*. This fungus has been reported to produce appressoria that are able to build an enormous turgor pressure, which makes it an attractive model to test these mechanosensors in. By making use of diverse mutants that are known to be affected in appressorial development or functioning, the authors convincingly show that the mechanosensor differentially response to diverse turgor pressures, and can be used as a read-out for such pressure. Interestingly, their study shows that membranes are perhaps not as homogenous as previously thought, and that spatial heterogeneity occurs.

Overall, the manuscript is very clear, well written, and convincing. The data that are produced are robust, and the conclusions justified based on the data that are presented. Admittedly, I read this paper mostly as a technical advance, as I feel that the advance in our understanding of the (infection)biology of *Magnaporthe* is limited, and there where such advance is provided (e.g. the heterogeneous nature of the appressorial membrane) the advance remains quite descriptive.

In their title, the authors refer to "Direct *measurement* of appressorium turgor", but after reading the manuscript I am not so sure whether this is indeed what the authors did. That is, I see mention of more and of less membrane tension, but this is not extrapolated to turgor measurements, leading to infernal of turgor pressure values. Can the authors comment on whether inferring such values is possible based on the currently presented technology? And if not, what would be required to be able to do so?

We thank the reviewer for their thoughtful comment. We have reworded the title of the manuscript in response to the reviewer concerns. A first attempt to quantification was made by the data presented in Extended Figure 2. The difficulty is that the probe response is sensitive to its local environment and a true calibration of the pressures can only be done in-situ. A speculative option to do so, which we now mention in the discussion section, but which is technically very challenging, would be to combine the FLIM read-out of the membrane tension probes with a direct measurement of the turgor using force-sensitive surfaces as reported by our group previously (Bronkhorst et al. *Nature Microbiology*). We have conducted preliminary experiments to do so, but have found that *Magnaporthe*, unlike the pathogen studied using this approach previously, does not readily form appressoria on these force-sensitive artificial surfaces, and surface engineering is required to offer *Magnaporthe* the cues it needs to form functional and invasive appressoria, a topic we will pursue in the future.

Also, I am wondering to what extent their observations are a direct consequence of membrane tensions versus degrees of melanin deposition that somehow affect the mechanosensors? I think in most of the mutants this is all correlated; less melanin, less turgor, but can the authors somehow uncouple these traits?

We thank the reviewer for this important comment. In order to decouple melanization from membrane tension, we decided to image *M. oryzae* germ tubes prior to incipient appressorium formation. Germ tubes are not melanised, in fact we know from microscopy and RNA-seq data, that melanin deposition begins after 4 hpi within the appressorium, with core enzymes required for DHN-melanin biosynthesis peaking in expression between 6-8 hpi. In the absence of melanin, the mechanoprobe showed local changes in tension at germ tube tips and also at points of curvature. We have included this result as a new Figure 4 in the paper.

We are grateful for all the comments and have responded to them as constructively as possible with new experimental work and expanded explanations. We believe the manuscript is much improved and thank the reviewers for the thoughtful remarks and suggestions.

Decision Letter, first revision:

Message: Our ref: NMICROBIOL-22082140A-Z

27th April 2023

Dear Nick,

Thank you for submitting your revised manuscript "Real-time visualization of appressorium turgor using a molecular mechanosensor in the rice blast fungus *Magnaporthe oryzae*" (NMICROBIOL-22082140A-Z). It has now been seen by the original referees and their comments are below. The reviewers find that the paper has improved in revision, and therefore we'll be happy in principle to publish it in Nature Microbiology, pending minor revisions to satisfy the referees' final requests and to comply with our editorial and formatting guidelines.

If the current version of your manuscript is in a PDF format, please email us a copy of the file in an editable format (Microsoft Word or LaTeX)-- we can not proceed with PDFs at this

1stage.

Thank you again for your interest in Nature Microbiology Please do not hesitate to contact me if you have any questions.

Sincerely,
[redacted]

Reviewer #1 (Remarks to the Author):

I thank the authors for adequately addressing my previous concerns.

Reviewer #2 (Remarks to the Author):

While the manuscript is improved I am still not wholly convinced by the arguments relating to novelty. I would also agree with the first two comments raised by Reviewer 3 and the arguments put forward by the authors to counter these are again not wholly convincing. I would argue that reduction of tension in a turgid protoplast would not require large amounts of lipids to be incorporated into the plasma membrane given that most of the tension would be borne by the cell wall. The relationship between membrane tension and turgor pressure is likely to be a complex one and this would raise concerns regarding how accurately the probe reflects turgor pressure. Given these concerns, despite the improvements I am still reluctant to recommend publication in Nature Microbiology.

Reviewer #3 (Remarks to the Author):

The authors have tried to address the points made by all reviewers in detail. I am mostly satisfied by the answers but a few concerns remain:

Regarding my original Point 1: I had asked the question why the authors think that the membrane is under tension when really the tensile stress caused by the internal pressure is being withheld by the cell wall rather than the membrane. The membrane is appressed against the wall (and thus experiences compressive force in normal direction), but I argued that under the time scales relevant here, from the presence of pressure does not necessarily result that the membrane proper is under tensile stress in its plane. In their rebuttal, the authors provide additional arguments for my point, apparently without realizing it. They state that upon enzymatic removal of the wall the cell would inflate to many times its original size. Given that the elasticity of plasma membrane material is only about 2%, a turgor driven inflation should be minimal and result in almost immediate failure and bursting. Instead, the authors speak of a substantial surface increase that

2confirms exactly what I hypothesized, that there is significant slack in the membrane that, upon removal of the spatial confinement, unfolds to allow for increase in cell volume. By consequence, in the confined state (with wall) the membrane would not have been under tension. The authors' argument about the mechanosensitive channels is more compelling although the exact mechanics of these channels is still an open question. I bring up the topic not at all to prevent next steps towards publication, but to make the authors reflect about the danger of continuing to propagate statements for which there is no real evidence yet. For example, in line 401 they speak of 'huge levels of tension [in the membrane]' but to my knowledge, no absolute value for the membrane tension in walled cells has been measured and given my arguments above, I don't actually think that the tension is so extremely high. This does not preclude changes in tension with varying turgor. I just don't think the absolute value of tension in the membrane is as high as the turgor values seem to make many people think. Conceptually, it doesn't need to be - the wall does the job.

The title: Even though the authors changed the title to remove the notion that they 'directly measure turgor', the revised title ('visualization of turgor') continues to be misleading. It should be 'visualization of membrane tension'. This distinction is important since the spatial heterogeneity of the signal across the surface of a cell is not caused by heterogeneity in local turgor, but reflects heterogeneity in the locally experienced tension.

Regarding my original Point 3: In response to my comment the authors have added information but did not explain how the ROI were chosen. On a circular object with changing width of the edge it remains unclear to me how the averaging was done to obtain a value for a given cell. For example in Fig. 5d, I presume that 'average fluorescence lifetime' means that one datapoint represents the averaged lifetime for a single cell and hence integrates the signal across the entire surface of a cell? If this is so, how was this done exactly? What do the ROIs look like? If my interpretation is wrong, please explain what a datapoint on the graph is. Also, I presume that R means replicate? Even if this is maybe explained in other figure legends, it should be mentioned here.

Reviewer #4 (Remarks to the Author):

My concerns have been appropriately addressed by the authors; I congratulate them with their nice study.

Author Rebuttal, first revision:

Please find below our responses to the comments of the reviewers following the second submission of the review process for Nature Microbiology. We have addressed the reviewer comments below in red, and are grateful to the editor and reviewers for accepting our manuscript for publication in Nature Microbiology.

3Reviewer #1:

Remarks to the Author:

I thank the authors for adequately addressing my previous concerns.

We thank the reviewer for their helpful suggestions and comments, which have contributed towards improving our manuscript for publication.

Reviewer #2:

Remarks to the Author:

While the manuscript is improved I am still not wholly convinced by the arguments relating to novelty. I would also agree with the first two comments raised by Reviewer 3 and the arguments put forward by the authors to counter these are again not wholly convincing. I would argue that reduction of tension in a turgid protoplast would not require large amounts of lipids to be incorporated into the plasma membrane given that most of the tension would be borne by the cell wall. The relationship between membrane tension and turgor pressure is likely to be a complex one and this would raise concerns regarding how accurately the probe reflects turgor pressure. Given these concerns, despite the improvements I am still reluctant to recommend publication in Nature Microbiology.

We respond to the comments of both Reviewer 2 and 3 below.

Reviewer #3:

Remarks to the Author:

The authors have tried to address the points made by all reviewers in detail. I am mostly satisfied by the answers but a few concerns remain:

Regarding my original Point 1: I had asked the question why the authors think that the membrane is under tension when really the tensile stress caused by the internal pressure is being withheld by the cell wall rather than the membrane. The membrane is appressed against the wall (and thus experiences compressive force in normal direction), but I argued that under the time scales relevant here, from the presence of pressure does not necessarily result that the membrane proper is under tensile stress in its plane. In their rebuttal, the authors provide additional arguments for my point, apparently without realizing it. They state that upon enzymatic removal of the wall the cell would inflate to many times its original size. Given that the elasticity of plasma membrane material is only about 2%, a turgor driven inflation should be minimal and result in almost immediate failure and bursting. Instead, the authors speak of a

4substantial surface increase that confirms exactly what I hypothesized, that there is significant slack in the membrane that, upon removal of the spatial confinement, unfolds to allow for increase in cell volume. By consequence, in the confined state (with wall) the membrane would not have been under tension. The authors' argument about the mechanosensitive channels is more compelling although the exact mechanics of these channels is still an open question.

We agree with reviewers 2 and 3 that the relationship between membrane tension and turgor pressure is complex, even more so given the results in our paper which reveal the emergence of strong lateral membrane inhomogeneities at later stages of appressorium development when turgor pressure is very high. The discussion of both reviewers is focused on the putative role of lipid biogenesis in relieving tensions, and while this is indeed an interesting discussion, we cannot but speculate about this point as we did not study it directly. Clearly, the reviewers can also only speculate on this point too, as there is no experimental evidence for their ideas.

The main point of our manuscript is that we provide evidence for a quantitative link between turgor and membrane tension in appressorium morphogenesis. The facts that support this are as follows:

1. Turgor is known to activate tension-activated ion channels, providing direct evidence for a link between turgor and membrane tension in the literature
2. Our data show that as appressorium development proceeds, which is known to involve the build-up of turgor pressure, that membrane tension increases.
3. Control experiments in which we counter turgor acting on the cell boundary (membrane & cell wall), by increasing the osmotic pressure in the medium, give a consistent result in the other direction (ie an easing of membrane tension)
4. Based on previous studies we know these probes are specifically localized in membranes and not the cell wall.

Based on these facts, and the absence of any arguments that counter this, the logical conclusion is that increased turgor during appressorium development indeed leads to increased membrane tension. That the quantitative link is complex is indeed also clear from our results. For example, the emergence of strong lateral inhomogeneities in the membrane hints at processes that couple wall and membrane mechanics, which remain elusive and which have not been remarked upon, or studied before (now mentioned on p.16 of Discussion).

On the role of lipid biosynthesis and the relaxation of tensions, we cannot do more than raise interesting points for future study. How the rates of lipid biogenesis and exocytosis to the plasma membrane, for example, compare to the rate of turgor generation (i.e. glycerol biosynthesis), in dictating membrane tension changes in time is wholly unknown. Our data

merely show that membrane tension increases, but not by what mechanism or whether there are suppressing-effects present that moderate membrane tension.

The link between membrane tension and turgor is largely understudied and our work provides new information to kick-start such studies. The fact that we cannot solve the entire puzzle at once is not surprising given the vast complexity of different kinetic processes, a variety of membrane-cell wall anchoring proteins involved, and the potential role of cytoskeletal networks that can also exert forces onto the plasma membrane from inside the cell, but in our opinion this does not take away from what we have achieved in this study. We have shown that membrane tension is associated with the extreme cellular turgor in an appressorium and provided a new, robust and quantitative method to study appressorium function

I bring up the topic not at all to prevent next steps towards publication, but to make the authors reflect about the danger of continuing to propagate statements for which there is no real evidence yet. For example, in line 401 they speak of 'huge levels of tension [in the membrane]' but to my knowledge, no absolute value for the membrane tension in walled cells has been measured and given my arguments above, I don't actually think that the tension is so extremely high. This does not preclude changes in tension with varying turgor. I just don't think the absolute value of tension in the membrane is as high as the turgor values seem to make many people think. Conceptually, it doesn't need to be - the wall does the job.

We thank the reviewer for their fair comment. We cannot attest to the magnitude of the tension, and it may not be as high. Can merely attest to the fact that there is tension, and that it is very heterogeneously distributed. We have adjusted these statements to reflect our observations more directly.

The title: Even though the authors changed the title to remove the notion that they 'directly measure turgor', the revised title ('visualization of turgor') continues to be misleading. It should be 'visualization of membrane tension'. This distinction is important since the spatial heterogeneity of the signal across the surface of a cell is not caused by heterogeneity in local turgor, but reflects heterogeneity in the locally experienced tension.

We thank the reviewer for this comment. We have changed the title to 'A molecular mechanosensor for real-time visualization of appressorium membrane tension in *Magnaporthe oryzae*'

Regarding my original Point 3: In response to my comment the authors have added information but did not explain how the ROI were chosen. On a circular object with changing width of the edge it remains unclear to me how the averaging was done to obtain a value for a given cell. For example in Fig. 5d, I presume that 'average fluorescence lifetime' means that one datapoint represents the averaged lifetime for a single cell and hence integrates the signal across the entire surface of a cell? If this is so, how was this done exactly? What do the ROIs look like? If my interpretation is wrong, please explain what a datapoint on the graph is. Also, I presume that R means replicate? Even if this is maybe explained in other figure legends, it should be mentioned here.

We thank the reviewer for this comment. To explain how the ROI was chosen for analysis, we have provided more explanation in the Materials and Methods section and have added an example of a typical ROI used for analysis as a new Extended Data Figure 1. This figure highlights the area of an appressorium used for analysis. We have also modified the figure legends to make this clear. To summarise, using SymPhoTime 64 software the ROI was drawn by hand in the 2D image which closely followed the membrane of the appressorium and did not include signal arising from any other part of the sample. The same software was used to fit the overall fluorescence decay curve of the ROI with a three-component exponential decay function. The fits were only deemed acceptable if the residuals were evenly distributed around zero and the χ^2 values were within the 0.70-1.30 range. Each datapoint in our analysis corresponds to the weighted average fluorescence lifetime obtained for the ROI of a single 2D image of an individual appressorium.

Reviewer #4:

Remarks to the Author:

My concerns have been appropriately addressed by the authors; I congratulate them with their nice study.

We thank the reviewer for their helpful suggestions and comments, which we feel has contributed towards improving our manuscript for publication.

Final Decision Letter:

Message 19th June 2023

:
Dear Nick,

I am pleased to accept your Article "A molecular mechanosensor for real-time visualization of appressorium membrane tension in *Magnaporthe oryzae*" for publication in Nature Microbiology. Thank you for having chosen to submit your work to us and many congratulations.

Acceptance of your manuscript is conditional on all authors' agreement with our publication policies (see <https://www.nature.com/nmicrobiol/editorial-policies>). In particular your manuscript must not be published elsewhere and there must be no announcement of the work to any media outlet until the publication date (the day on which it is uploaded onto our website).

Please note that *Nature Microbiology* is a Transformative Journal (TJ). Authors may publish their research with us through the traditional subscription access route or make their paper immediately open access through payment of an article-processing charge (APC). Authors will not be required to make a final decision about access to their article until it has been accepted. [Find out more about Transformative Journals](https://www.springernature.com/gp/open-research/transformative-journals)

Authors may need to take specific actions to achieve [open access](https://www.springernature.com/gp/open-research/funding/policy-)

8compliance-faqs"> compliance with funder and institutional open access mandates.

If your research is supported by a funder that requires immediate open access (e.g. according to [Plan S principles](https://www.springernature.com/gp/open-research/plan-s-compliance)) then you should select the gold OA route, and we will direct you to the compliant route where possible. For authors selecting the subscription publication route, the journal's standard licensing terms will need to be accepted, including [self-archiving policies](https://www.nature.com/nature-portfolio/editorial-policies/self-archiving-and-license-to-publish). Those licensing terms will supersede any other terms that the author or any third party may assert apply to any version of the manuscript.

With kind regards,

[redacted]